# Efficient Multi-Agent Reasoning via Confidence-Guided Adaptive Debate

**Seungdong Yoa** [1]  **Ye Seul Sim** [1]  **Suhee Yoon** [1]  **Sanghyu Yoon** [1]  **Dongmin Kim** [1]  **Soonyoung Lee** [1]
**Bumsoo Kim**[†, 2]  **Junhyun Lee**[†, 3 4]

## Abstract

Multi-agent debate has shown promise for improving the reasoning of large language models, yet recent theory suggests its benefits are highly regime-dependent. While interaction can amplify informative signals under corrective conditions, symmetric debate dynamics are neutral in expectation, often making majority voting preferable. We reconcile these views by arguing that debate is effective only when invoked at the right time and with appropriate structure. Based on this insight, we propose LASE: Leader-Adaptive Structured Engagement, a leader-centric multi-agent debate framework that selectively engages interaction only in non-neutral regimes. LASE introduces an asymmetric leader–supporter architecture that enables directed information flow and selective signal amplification, while defaulting to simple aggregation otherwise. Experiments across diverse reasoning benchmarks show that LASE achieves multi-agent-level performance with near single-agent token cost, substantially improving efficiency over static debate and voting baselines.

## 1. Introduction

Multi-agent debate has emerged as a promising paradigm for improving the reasoning performance of large language models (LLMs), motivated by the intuition that interaction among agents can correct individual errors and refine collective beliefs. In its canonical form, multi-agent debate combines two ingredients: iterative inter-agent interaction and a final aggregation mechanism, often majority vote. While empirically successful in some settings, recent theoretical

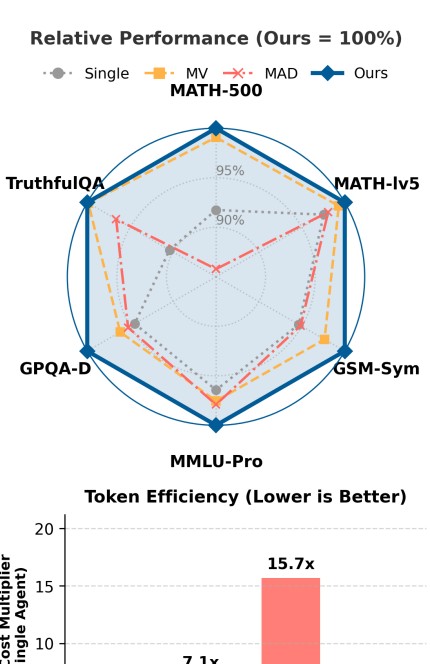

*Figure 1.* **Performance and Efficiency Overview.** (Top) Relative performance comparison (Ours=100%). Ours consistently outperforms baselines across all tasks. (Bottom) **Token cost** multipliers vs. Single Agent. Ours incurs minimal overhead compared to the computationally expensive MAD.

work has revealed a nuanced and seemingly contradictory picture of *when and why* debate is effective.

Two recent lines of work offer complementary theoretical views of multi-agent debate. Hu et al. (2025) model debate as Bayesian inference over latent concepts, showing that when agent responses are conditionally independent given an unobserved concept, interaction corresponds to posterior updating and can amplify a correct seed, yielding strictly improved expected correctness and, under mild conditions, outperforming majority vote. In contrast, Choi et al. (2025) analyze debate in a discrete-choice Bayesian framework with Dirichlet-distributed beliefs updated via neighbor counts and show that, under symmetry assumptions equating agents' beliefs with their neighbors' average,

[1]LG AI Research, Seoul, South Korea [2]School of Computer Science and Engineering, Chung-Ang University, Seoul, South Korea [3]Division of Computer Engineering, Hankuk University of Foreign Studies, Yongin, South Korea [4]Noah's Farm. Correspondence to: Bumsoo Kim <bumsoo@cau.ac.kr>, Junhyun Lee <junhyun.lee@hufs.ac.kr>.

debate dynamics form a martingale that preserves expected correctness without inducing positive drift.

Taken together, these theories point to a shared conclusion: **debate is effective only in specific regimes**. When interaction introduces meaningful asymmetry—such as the presence of a correct or highly informative response—debate can act as a corrective mechanism that amplifies useful signals. In contrast, when agents are symmetric and information flow is unconstrained, debate primarily redistributes existing information and yields little benefit beyond ensembling. Crucially, the distinction between these regimes is not intrinsic to the agents themselves, but to the *timing* and *structure* of interaction.

This perspective reframes the central challenge of multi-agent debate. Rather than asking whether debate is beneficial in general, the more fundamental question is when debate should be invoked and how it should be structured to exploit corrective conditions while avoiding neutral ones. Existing frameworks typically apply debate uniformly across instances, implicitly conflating regimes in which interaction is theoretically advantageous with those in which it is not. As a result, debate is often employed even when its expected contribution is minimal, incurring computational overhead without corresponding gains in accuracy.

In this work, we build on these theoretical insights to develop LASE: Leader-Adaptive Structured Engagement, a leader-centric multi-agent debate framework that selectively invokes additional interactions only when a *non-neutral* regime is likely. Our approach explicitly targets instances where interaction is likely to be corrective and structures information flow to selectively amplify informative signals, while defaulting to simpler aggregation when debate is expected to be neutral. By aligning the use of debate with its underlying theoretical regimes, our framework reconciles prior results and provides a principled foundation for more effective and efficient multi-agent reasoning. We propose a leader–supporter multi-agent framework in which a designated leader agent maintains a coherent global reasoning trajectory, while supporter agents provide auxiliary signals that influence, but do not symmetrically override, the leader's decision process. This asymmetric interaction breaks the symmetry inherent in prior MAD systems and introduces a directed flow of information, enabling supporter feedback to function as a corrective bias rather than a redundant exchange.

LASE can be viewed as imposing a structured update rule over the debate process: the leader aggregates supporter inputs under a constrained decision policy, transforming debate from an expectation-preserving interaction into a biased yet controlled reasoning operator. Empirically, we show that this design recovers the benefits of collective reasoning while avoiding the inefficiencies of unstructured debate, resulting in more robust and scalable multi-agent inference.

We evaluate LASE on a suite of challenging reasoning benchmarks spanning mathematical, symbolic, and knowledge-intensive tasks. Experimental results show that LASE achieves multi-agent-level performance with near single-agent token cost, substantially reducing token usage compared to static multi-agent debate and majority-voting baselines. These findings highlight confidence-guided test-time adaptation as a promising direction for scalable and efficient multi-agent reasoning.

## 2. Preliminaries

We focus on the problem of test-time multi-agent inference, where a system of agents collaborates to solve tasks given input text. In this section, we formalize the problem setup and review two complementary theoretical perspectives on debate dynamics. We then present a unified view that motivates our proposed structured framework.

### 2.1. Problem Formulation

Let $x$ denote an input instance (e.g., a question or problem context) and $y \in \mathcal{Y}$ be the corresponding ground-truth output. We consider a system of $n$ agents indexed by $i \in \{1, \ldots, n\}$. At each debate round $t \in \{0, \ldots, T\}$, agent $i$ outputs a response $z_i^{(t)}$ based on the input and the debate history $Z^{0:t-1}$. Let $Z^{(t)} = (z_1^{(t)}, \ldots, z_n^{(t)})$ denote the collection of responses at round $t$. A deterministic extractor $e(\cdot)$ maps a response to a discrete answer $\hat{y}_i^{(t)} := e(z_i^{(t)})$.

We compare two primary aggregation strategies. The first is *Majority Voting* (MV), defined as $\mathrm{MV}(Z^{(t)}) = \mathrm{mode}(\{\hat{y}_i^{(t)}\}_{i=1}^n)$. Under classical assumptions where individual accuracy satisfies $p > 0.5$, majority voting provably improves correctness as $n$ increases. The second is the *Debated Outcome*, denoted as $D(Z^{0:T})$, which represents the final consensus derived after iterative interaction among agents.

### 2.2. Prior Theoretical Perspectives: When Debate Helps vs. When Vote Suffices

Two recent lines of work provide complementary (and seemingly opposing) theoretical accounts of multi-agent debate. One formalizes debate as Bayesian inference over latent concepts and proves conditions under which interaction improves over majority vote (Hu et al., 2025). The other models debate as Bayesian updating in a discrete-choice (Dirichlet-multinomial) setting and shows that, under symmetry, debate can be *neutral* in expectation (a martingale), explaining why voting may dominate in practice (Choi et al., 2025).

**Latent-concept Bayes view (Debate can help).** Hu et al. (2025) introduce a latent concept $\theta \in \Theta$ governing agent responses and assume conditional independence given $\theta$. For simplicity, we suppress agent parameters and absorb conditioning on the input and debate history into the latent concept.

**Assumption 2.1** (Conditional independence on latent concepts; Hu et al. (2025)). *Conditioned on $\theta$, an agent's next response distribution is independent of both the debate history and the input:*

$$P(z_i^{(t+1)} \mid \theta, x, Z^{0:t}) = P(z_i^{(t+1)} \mid \theta). \quad (1)$$

Under this assumption, each agent's response distribution can be written as a Bayesian mixture over $\theta$ (posterior updating using peers' outputs).

**Lemma 2.2** (Bayesian mixture / posterior update; Hu et al. (2025)). *Under Assumption 2.1, for agent $i$,*

$$P(z_i^{(t+1)} \mid x, Z^{0:t}) \propto \quad (2)$$

$$\sum_{\theta \in \Theta} P(z_i^{(t+1)} \mid \theta) \, P(\theta) \, P(x \mid \theta) \prod_{j=1}^{n} P(z_j^{(t)} \mid \theta).$$

A central sufficient condition is the presence of at least one response strongly consistent with the true concept (a correct seed).

**Assumption 2.3** (Initial correct seed; Hu et al. (2025)). *At $t = 0$, at least one agent produces a response that is more compatible with the true concept $\theta^\star$ than with any $\theta \neq \theta^\star$.*

With a seed, iterative interaction can amplify correctness.

**Theorem 2.4** (Consistent response amplification; Hu et al. (2025)). *If the round-$t$ pool contains a response strongly consistent with $\theta^\star$, then the expected correctness at round $t+1$ is strictly higher than in the case where no such response exists.*

Moreover, under additional regularity conditions, the final debated outcome can exceed the initial majority vote.

**Theorem 2.5** (Debate can outperform majority vote; Hu et al. (2025)). *Under Assumptions 2.1-2.3 and mild regularity conditions, $P(D(Z^{0:T}) = y) > P(MV(Z^{(0)}) = y)$.*

**Dirichlet-multinomial discrete-choice view (Debate can be neutral).** Choi et al. (2025) model each agent's belief over $K$ discrete options as a Dirichlet random variable updated by neighbor counts. Let $\alpha_{i,t} \in \mathbb{R}_+^K$ be the Dirichlet parameters for agent $i$ at round $t$, and $c_{i,t}$ be the count vector of neighbors' responses.

**Lemma 2.6** (Dirichlet conjugacy for neighbor updates; Choi et al. (2025)). *With a prior $\theta_{i,t} \sim \text{Dirichlet}(\alpha_{i,t-1})$, the posterior after observing neighbor counts $c_{i,t}$ remains Dirichlet with updated parameters $\alpha_{i,t} = \alpha_{i,t-1} + c_{i,t}$.*

Let $p_{i,t}$ denote agent $i$'s posterior mean probability assigned to the correct option derived from $\alpha_{i,t}$. Under a symmetry condition equating an agent's belief with the average belief of its neighbors, the belief trajectory is a martingale.

**Theorem 2.7** (Martingale neutrality under symmetry; Choi et al. (2025)). *If at round $t - 1$ agent $i$'s belief equals the average of its neighbors' beliefs, then $\mathbb{E}[p_{i,t} \mid \alpha_{i,t-1}] = p_{i,t-1}$.*

This result implies that in homogeneous fully-connected settings, debate alone does not induce positive expected drift toward correctness. In such settings, the debate dynamics are approximately *expectation-preserving*: $\mathbb{E}[\bar{b}_{t+1} \mid s_t] \approx \bar{b}_t$. That is, while debate may affect variance or inter-agent correlation, it does not systematically increase expected correctness, providing a mathematical explanation for empirical observations where unstructured debate offers limited gains over majority voting.

## 2.3. Unified Interpretation: Breaking Symmetry via Structured Debate

The two theories are compatible: Theorem 2.7 characterizes a *neutral regime* where symmetric information flow yields zero expected improvement, rationalizing why majority vote can dominate empirically. In contrast, Theorems 2.4-2.5 identify a *corrective regime* in which asymmetric likelihood signals—most simply, the existence of at least one correct seed—create positive drift via posterior amplification.

This reconciliation motivates methods that target interaction to instances where corrective conditions are likely to hold. To reliably outperform expectation-preserving interactions, we must introduce *structure* that breaks symmetry and induces a biased update toward correctness. We formalize this goal as:

$$\bar{b}_{t+1} = \bar{b}_t + \Delta_t, \quad \text{where} \quad \mathbb{E}[\Delta_t \mid Z^{0:t}] > 0. \quad (3)$$

Here, $\bar{b}_t = \frac{1}{n} \sum_{i=1}^{n} p_{i,t}$ represents the **average belief** of the agent population in the correct answer.

To achieve $\Delta_t > 0$, we propose a **Structured Debate** protocol via role differentiation. Let $\mathcal{S} \subset \{1, \ldots, n\}$ denote a set of *Supporters* who generate auxiliary signals $u_j$ (e.g., critiques or verifications). By designating a *Leader* ($L$) to synthesize auxiliary signals from *Supporters* ($\mathcal{S}$), we enforce a directed aggregation function $S_L$:

$$\hat{y} = \arg\max_{y' \in \mathcal{Y}} S_L(x, \{u_j\}_{j \in \mathcal{S}}). \quad (4)$$

This asymmetric interaction transforms debate from a neutral information-mixing process into a controlled reasoning operator, forcing the system into the corrective regime.

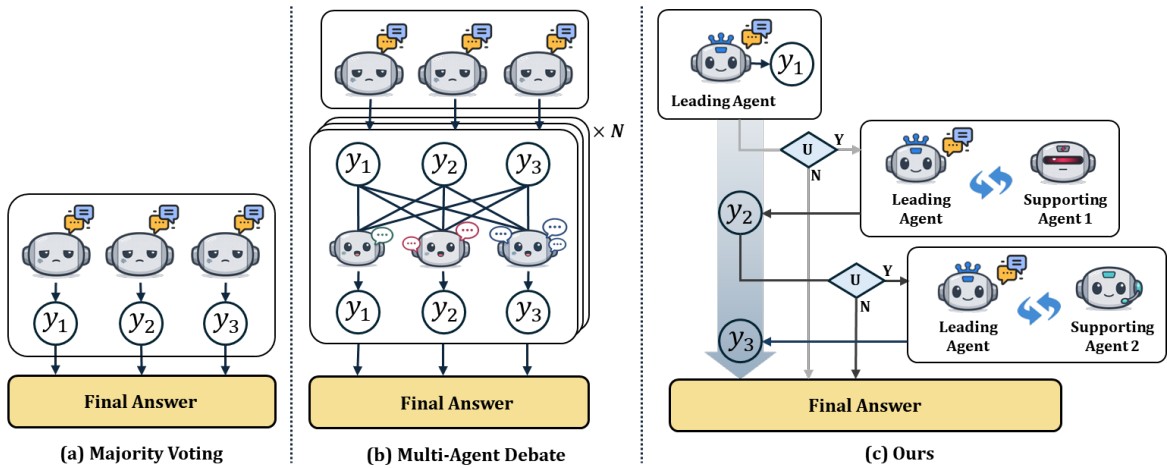

*Figure 2.* Overall architectures of (a) Majority Voting, (b) Multi-Agent Debate and (c) Ours. (a) aggregates independent outputs, (b) applies uniform interaction while (c) selectively invokes structured, leader-centric interaction in corrective regimes where $U = 1$ in Eq. (7).

## 3. Method

We propose **LASE**, a leader-centric test-time inference framework that selectively invokes interaction only when a *corrective* regime is likely, while avoiding interaction in regimes where debate is expected to be neutral in expectation (cf. Theorem 2.7) or dominated by voting. Throughout, we keep the notation from Section 2.2. Without loss of generality, let the *leader* be agent $L := 1$, and let $\mathcal{P} := \{2, \ldots, n\}$ denote the set of potential peers. All proofs of our theoretical claims are provided in the Appendix.

**Token-level predictive distributions.** For an agent $i$ at round $t$, write its next-token distribution as $p_i^{(t)}(\cdot \mid x, Z^{0:t-1})$. For a generated response $z_i^{(t)} = (w_1, \ldots, w_m)$, define token log-probabilities $\ell_k := \log p_i^{(t)}(w_k \mid x, Z^{0:t-1}, w_{<k})$ and token entropies $h_k := H\left(p_i^{(t)}(\cdot \mid x, Z^{0:t-1}, w_{<k})\right)$. Let $\mathcal{T}(z_i^{(t)}) \subseteq \{1, \ldots, m\}$ denote the token indices corresponding to the extracted answer span used by $e(\cdot)$.

**Definition 3.1** (Confidence proxies). For the leader's response $z_L^{(t)}$, define

$$\text{LP}_{\min}(z_L^{(t)}) := \min_{k \in \mathcal{T}(z_L^{(t)})} \ell_k, \tag{5}$$

$$\text{H}_{\max}(z_L^{(t)}) := \max_{k \in \mathcal{T}(z_L^{(t)})} h_k, \tag{6}$$

and the (thresholded) uncertainty event

$$U(z_L^{(t)}) := \mathbf{1}\left\{\text{LP}_{\min}(z_L^{(t)}) < \tau_{\text{lp}} \ \vee \ \text{H}_{\max}(z_L^{(t)}) > \tau_{\text{h}}\right\}. \tag{7}$$

**LASE framework.** LASE begins with a leader-only attempt:

$$z_L^{(0)} \sim P(z_L^{(0)} \mid x), \qquad \hat{y}_{\text{solo}} := e(z_L^{(0)}). \tag{8}$$

If $U(z_L^{(0)}) = 0$, LASE stops and outputs $\hat{y}_{\text{solo}}$. Otherwise, LASE performs staged escalation with at most *one* peer at a time: (i) a low-cost *adversarial* check, then (ii) a peer *support* interaction, and optionally (iii) an *expert* escalation. Formally, LASE selects (possibly different) indices $A, S, E \in \mathcal{P}$ and generates

$$z_A^{(0)} \sim P(z_A^{(0)} \mid x, z_L^{(0)}), \tag{9}$$

$$z_L^{(1)} \sim P(z_L^{(1)} \mid x, z_L^{(0)}, z_A^{(0)}), \tag{10}$$

$$z_S^{(0)} \sim P(z_S^{(0)} \mid x, z_L^{(1)}), \tag{11}$$

$$z_L^{(2)} \sim P(z_L^{(2)} \mid x, z_L^{(1)}, z_S^{(0)}), \tag{12}$$

with early stopping whenever $U(z_L^{(t)}) = 0$, and final output $\hat{y}_{\text{LASE}} := e(z_L^{(t^\star)})$ at the stopping round $t^\star$. This yields an interaction graph that is a directed star centered at the leader (one peer at a time), rather than all-to-all interaction.

### 3.1. Why Selective Interaction: Reconciling Neutral vs. Corrective Regimes

Theorem 2.7 characterizes a *neutral* regime under symmetry, where homogeneous fully-connected updating yields no positive expected drift toward correctness. LASE departs from this regime in two ways: (a) *topology* (leader-centric, not fully-connected), and (b) *conditioning* (interaction is invoked only on the event $U = 1$).

**Proposition 3.2** (LASE breaks symmetry conditions underlying martingale neutrality). *Consider a discrete-choice*

*abstraction of the type used in Choi et al. (2025), with posterior-mean beliefs $p_{i,t}$ updated by neighbor information. If interaction is restricted to a directed star centered at $L$ (i.e., only $L$ receives peer messages) and activated only on an event $U(z_L^{(t)}) = 1$ measurable w.r.t. $L$'s transcript, then the symmetry condition "$p_{i,t-1}$ equals the average of its neighbors' beliefs" required by Theorem 2.7 generally fails for the leader's update. Consequently, martingale neutrality does not apply to the leader's belief trajectory under LASE.*

In the latent-concept Bayes view of Hu et al. (2025), interaction can be *corrective* when a strong seed exists (Assumption 2.3), enabling amplification (Theorem 2.4) and eventual improvements over majority vote (Theorem 2.5). LASE aims to concentrate interaction budget on instances where such corrective conditions are likely.

## 3.2. Selective Debate Improves the Performance-Cost Tradeoff

We formalize LASE as a test-time resource allocation policy and show that, under mild conditions, selective interaction yields a Pareto improvement over static debate (always interact) and static vote (always aggregate).

**Accuracy and token cost.** Let $\hat{y}_\pi$ be the output of a policy $\pi$ (e.g., LASE, static debate, majority vote), and define its expected accuracy

$$\mathrm{Acc}(\pi) := \mathbb{P}(\hat{y}_\pi = y). \quad (13)$$

Let $\kappa(z)$ denote the number of generated tokens in $z$ and let $c_i > 0$ be a per-token cost for agent $i$ (capturing latency/price/model size). Define the total expected token cost

$$\mathrm{Cost}(\pi) := \mathbb{E}\Big[ \sum_{\text{calls } (i,t) \text{ made by } \pi} c_i \, \kappa\big(z_i^{(t)}\big) \Big]. \quad (14)$$

**Key assumptions (calibration & conditional helpfulness).** We use uncertainty scores to gate interaction.

**Assumption 3.3** (Monotone calibration of uncertainty). There exists a scalar score $s(z_L^{(t)})$ that is nondecreasing in uncertainty (e.g., a function of $\mathrm{LP}_{\min}$ and $\mathrm{H}_{\max}$) such that $q(u) := \mathbb{P}(e(z_L^{(0)}) = y \mid s(z_L^{(0)}) = u)$ is nonincreasing in $u$.

We provide empirical evidence to validate Assumption 3.3, as shown in Figure 3.

Let $Q(x, z_L^{(0)}) \subseteq \mathcal{P}$ denote the set of peers queried by LASE on input $x$ when $U(z_L^{(0)}) = 1$. Define the queried-seed event

$$\mathcal{S}_{\text{seed}} := \{\exists j \in Q(x, z_L^{(0)}) : z_j^{(0)} \text{ is a correct seed}\}, \quad (15)$$

where "correct seed" is understood in the sense of Assumption 2.3. Let $\mathcal{S}_{\text{seed}}^c$ denote its complement, i.e., the event that no queried peer provides a correct seed.

**Assumption 3.4** (Conditional seed prevalence on uncertain instances). There exists $\rho \in (0, 1]$ such that, conditional on the leader being uncertain, the peer set actually queried by LASE contains at least one correct seed with probability at least $\rho$:

$$\mathbb{P}\big(\mathcal{S}_{\text{seed}} \mid U(z_L^{(0)}) = 1\big) \geq \rho. \quad (16)$$

The probability is taken over the data distribution, model stochasticity, and any randomness in the peer-selection rule.

**Assumption 3.5** (Conditional amplification under a seed). There exists $\delta > 0$ such that whenever the leader conditions on at least one correct-seed peer response (as in Lemma 2.2), the probability of a correct extracted answer strictly increases by at least $\delta$:

$$\mathbb{P}\big(e(z_L^{(1)}) = y \mid \text{a correct seed is observed}\big)$$
$$\geq \mathbb{P}\big(e(z_L^{(0)}) = y\big) + \delta, \quad (17)$$

where the update from $z_L^{(0)}$ to $z_L^{(1)}$ is any single-step interaction consistent with the Bayesian mixture view (Lemma 2.2).

Assumptions 3.4 - 3.5 are natural in the corrective regime of Hu et al. (2025): when a seed appears, posterior mass on $\theta^\star$ increases and can be amplified across refinement. Assumption 3.3 is empirically testable (e.g., via reliability plots of correctness vs. $s(\cdot)$).

**Expected gain from selective interaction.** Let $\pi_{\text{solo}}$ be the leader-only policy (stop after $z_L^{(0)}$), $\pi_{\text{full}}$ be a static debate policy that always performs at least one peer interaction, and $\pi_{\text{LASE}}$ be the threshold policy that performs interaction iff $U(z_L^{(0)}) = 1$.

**Assumption 3.6** (Conservativeness when no seed). When no correct seed is observed on uncertain instances, LASE is no worse in expectation than keeping the leader-only answer:

$$\mathbb{P}(\hat{y}_{\text{LASE}} = y \mid U = 1, \mathcal{S}_{\text{seed}}^c) \geq \mathbb{P}(\hat{y}_{\text{solo}} = y \mid U = 1). \quad (18)$$

**Theorem 3.7** (Selective debate yields positive expected improvement). *Under Assumptions 3.4–3.6,*

$$\mathrm{Acc}(\pi_{LASE}) \geq \mathrm{Acc}(\pi_{solo}) + \mathbb{P}\big(U(z_L^{(0)}) = 1\big) \cdot \rho\delta. \quad (19)$$

**Cost reduction from leader-centric gating and 1:1 topology.** Let $C_L := \mathbb{E}[c_L \kappa(z_L^{(0)})]$ and define $C_{\text{peer}}$ as the expected incremental cost of one peer interaction (including the peer message and the leader revision). Then

$$\mathrm{Cost}(\pi_{\text{LASE}}) = C_L + \mathbb{P}(U = 1) \cdot C_{\text{peer}}$$
$$\leq C_L + C_{\text{peer}} = \mathrm{Cost}(\pi_{\text{full}}), \quad (20)$$

with strict inequality whenever $\mathbb{P}(U = 1) < 1$.

Moreover, LASE's 1:1 topology reduces *context integration* cost relative to all-to-all debate.

**Proposition 3.8** (Interaction-topology cost scaling). *Consider $T$ rounds and $n$ agents. In an all-to-all debate where each agent conditions on every other agent's latest message each round, the number of cross-agent context inclusions scales as $\Theta(Tn^2)$. In contrast, a leader-centric 1:1 scheme that invokes at most one peer per escalation round scales as $\mathcal{O}(T)$ inclusions (independent of $n$), yielding an $\Omega(n^2)$ reduction in cross-agent context traffic at fixed $T$.*

**Pareto improvement.** Combining Theorem 3.7 with the cost identity above yields the central tradeoff: LASE attains nontrivial accuracy gains over leader-only inference while spending interaction budget only on uncertain instances, and it reduces cost relative to static debate whenever easy instances have non-negligible mass.

**Corollary 3.9** (Pareto frontier improvement). *Assume $\mathbb{P}(U(z_L^{(0)}) = 1) \in (0, 1)$ and Assumptions 3.4–3.6. Then $\pi_{LASE}$ strictly improves upon static debate $\pi_{full}$ in expected cost, while maintaining a positive expected accuracy gain over leader-only inference. In particular, for any fixed budget $B$ such that $\mathrm{Cost}(\pi_{LASE}) \leq B < \mathrm{Cost}(\pi_{full})$, LASE achieves an accuracy unattainable by always-debate under budget $B$, and (by Theorem 3.7) exceeds leader-only accuracy by at least $\mathbb{P}(U = 1)\rho\delta$.*

**Remark (compatibility with voting).** LASE is orthogonal to the aggregation rule used when peers are queried. For example, one may replace the single peer in the support stage by a small set of peers and use $\mathrm{MV}(\cdot)$ to propose a candidate answer, while preserving the same uncertainty-gated structure. Such hybrids correspond to using vote as an additional (possibly biased) signal to increase the conditional seed probability $\rho$ in Assumption 3.4, aligning with the "corrective" regime of Hu et al. (2025) while still avoiding the neutral regime highlighted by Choi et al. (2025).

## 4. Experiments

### 4.1. Experimental Setup

**Models and Baselines.** We evaluate our framework using two widely used general-purpose LLMs: **Gemini-2.0-Flash** [Google] and **GPT-4.1-mini** [OpenAI]. These models serve as the backbone for our experiments to validate the efficiency of the proposed framework. We compare LASE against three baselines: 1) Single Agent (standard zero-shot generation), 2) Majority Vote (MV), which samples $N = 7$ independent responses and applies majority voting, and 3) Multi-Agent Debate (MAD), where $N = 3$ agents engage in a multi-turn debate for 2 rounds followed by majority

voting.

**Datasets.** We evaluate on six benchmarks spanning mathematical reasoning and general knowledge. For reasoning, we use MATH-500 (Lightman et al., 2023), a custom MATH-lv5 subset ($N = 1,324$) containing only the highest-difficulty problems (Hendrycks et al., 2021), and GSM-Symbolic (Mirzadeh et al., 2024) ($N = 500$) to test robustness against memorization. For knowledge and safety, we employ MMLU-Pro (Wang et al., 2024a) ($N = 504$), GPQA-Diamond (Rein et al., 2024) ($N = 198$), and TruthfulQA-MC (Lin et al., 2022) ($N = 817$).

### 4.2. Main Results

**Superior Reasoning Accuracy.** As shown in Table 1, LASE consistently achieves state-of-the-art performance, outperforming computationally expensive ensembles (MV and MAD) on both Gemini-2.0-Flash and GPT-4.1-mini. Notably, on GPQA-Diamond, LASE reaches 65.2% accuracy, surpassing MV (62.7%) and MAD (62.1%), while also establishing a clear lead on MATH-500 with 87.2%.

**Order-of-Magnitude Efficiency Gain.** LASE achieves these gains with drastically reduced overhead, bridging the gap between multi-agent intelligence and single-agent cost. Our method reduces average token usage by over $10\times$ compared to MAD. Specifically, on GSM-Symbolic, LASE consumes only 0.4k tokens versus MAD's 5.6k (a $14\times$ reduction), remaining comparable to the single-agent baseline. Even on complex tasks like MATH-500, our cost (1.1k) is a fraction of MAD (11.8k) and MV (5.2k).

### 4.3. Compatibility with Majority Voting

A common question in multi-agent reasoning is how the proposed framework relates to simple ensemble methods like Majority Voting (MV). We clarify that our framework is not a mutually exclusive alternative to MV, but rather an orthogonal framework that enhances the reasoning quality of each individual path. To demonstrate this, we conducted an experiment where we run our method in parallel ($N = 5$) and aggregate the final answers via majority voting (denoted as **Ours-Majority Vote**). As shown in Table 2, Ours-Majority Vote further boosts performance by +0.8% compared to our single-path performance. Crucially, because our framework is highly token-efficient (invoking agents only when necessary), running ours 5 times consumes comparable tokens to the standard MV ($N = 7$). However, Ours-Majority Vote significantly outperforms MV. This suggests that ours serves as a "better voter": by filtering out errors within each reasoning path via adaptive debate, the final ensemble becomes more robust than simply aggregating raw, unverified generations. Thus, ours can be flexibly scaled up with MV for scenarios demanding maximum accuracy.

*Table 1.* **Main Results on Reasoning and Knowledge Benchmarks.** We compare ours with Single Agent, Majority Vote (MV), and Multi-Agent Debate (MAD) baselines. **Ours** consistently achieves multi-agent level accuracy while maintaining near single-agent token efficiency. Best results are highlighted in **bold** (Accuracy: across all methods; Tokens: among multi-agent frameworks).

| Model | Metric | MATH-500 (Math) | | | | MATH-lv5 (Hard Math) | | | | GSM-Symbolic (Robustness) | | | |
|---|---|---|---|---|---|---|---|---|---|---|---|---|---|
| | | Single | MV | MAD | **Ours** | Single | MV | MAD | **Ours** | Single | MV | MAD | **Ours** |
| **Gemini 2.0-Flash** | Accuracy (↑) | 80.0 | 86.4 | 74.8 | **87.2** | 55.7 | 56.7 | 56.0 | **57.1** | 88.4 | 91.2 | 88.6 | **93.4** |
| | Avg. Tokens (↓) | 0.7k | 5.2k | 11.8k | **1.1k** | 1.0k | 7.4k | 20.3k | **1.8k** | 0.4k | 2.6k | 5.6k | **0.4k** |
| | Total Tokens (↓) | 0.4M | 2.6M | 5.9M | **0.5M** | 1.3M | 9.7M | 26.9M | **2.4M** | 0.2M | 1.3M | 2.8M | **0.2M** |
| **GPT 4.1-mini** | Accuracy (↑) | 73.0 | 73.8 | 73.8 | **74.2** | 79.8 | 84.7 | 86.4 | **86.6** | 94.4 | **94.8** | 93.2 | **94.8** |
| | Avg. Tokens (↓) | 1.0k | 7.3k | 16.1k | **1.8k** | 1.6k | 11.7k | 29.1k | **2.6k** | 0.3k | 2.4k | 5.9k | **0.5k** |
| | Total Tokens (↓) | 0.5M | 3.6M | 8.0M | **0.9M** | 2.2M | 15.4M | 38.6M | **3.5M** | 0.2M | 1.2M | 2.9M | **0.2M** |

| Model | Metric | MMLU-Pro (Knowledge) | | | | GPQA-Diamond (Science) | | | | TruthfulQA (Safety) | | | |
|---|---|---|---|---|---|---|---|---|---|---|---|---|---|
| | | Single | MV | MAD | **Ours** | Single | MV | MAD | **Ours** | Single | MV | MAD | **Ours** |
| **Gemini 2.0-Flash** | Accuracy (↑) | 78.4 | 79.3 | 79.6 | **81.3** | 61.6 | 62.7 | 62.1 | **65.2** | 72.6 | 80.2 | 77.6 | **80.3** |
| | Avg. Tokens (↓) | 0.8k | 5.6k | 10.3k | **0.8k** | 1.0k | 7.5k | 17.2k | **1.2k** | 0.4k | 2.7k | 5.2k | **0.4k** |
| | Total Tokens (↓) | 0.4M | 2.8M | 5.2M | **0.4M** | 0.2M | 1.5M | 3.4M | **0.2M** | 0.3M | 2.2M | 4.3M | **0.3M** |
| **GPT 4.1-mini** | Accuracy (↑) | 79.4 | 81.5 | 81.7 | **81.8** | 67.7 | 71.2 | 72.7 | **72.9** | 79.9 | 80.9 | 80.8 | **81.1** |
| | Avg. Tokens (↓) | 1.0k | 6.8k | 11.7k | **1.0k** | 1.4k | 10.3k | 19.3k | **1.5k** | 0.5k | 3.4k | 5.7k | **0.5k** |
| | Total Tokens (↓) | 0.5M | 3.4M | 5.9M | **0.5M** | 0.3M | 2.0M | 3.8M | **0.3M** | 0.4M | 2.8M | 4.7M | **0.4M** |

*Table 2.* **Compatibility with Majority Voting (MATH-500).** comparison of standard MV ($N = 7$) vs. our ensemble variant (**Ours-Majority Vote**, $N = 5$).

| METHOD | $N$ | TOKENS | ACC. | $\Delta$ |
|---|---|---|---|---|
| MV | 7 | 5.2K | 86.4 | - |
| **OURS (SINGLE)** | 1 | **1.1K** | 87.2 | +0.8 |
| **OURS (MAJORITY VOTE)** | 5 | 5.4K | **88.0** | +1.6 |

## 4.4. Calibration Analysis: Validating Confidence Proxies

A core premise of our method is that the model's internal uncertainty signals can serve as reliable proxies for determining when external collaboration is necessary. To ensure statistical robustness and model-agnostic validity, we aggregated results from two independent runs across both Gemini-2.0-Flash and GPT-4.1-mini on the MATH-500 dataset, resulting in a total of **2,000 sample points** for this analysis. Figure 3 visualizes the relationship between our gating metrics and ground-truth accuracy based on this aggregated data.

**Strong Correlation with Performance.** As observed in Figure 3a, **Min Logprob** exhibits a strong positive correlation with accuracy. High-confidence generations ($\geq 0.99$) achieve 85.5% accuracy, whereas low-confidence outputs ($< 0.50$) drop to 50.9%, validating Min Logprob as an effective proxy for *token-level certainty*. Conversely, Figure 3b shows that **Entropy** is inversely correlated with performance. Notably, when entropy exceeds 1.0 (indicating a flat distribution or "ignorance"), accuracy plummets to 30.4%. This confirms that Entropy effectively captures *distributional ambiguity*.

*Table 3.* **Robustness to Agent Permutation (MATH-500).** We report three independent runs where the supporting agent order was randomly shuffled for each query. The low standard deviation (0.31) and the minimal gap ($\Delta 0.3\%$) from our main reported score confirm the stability of our framework.

| Setting | Accuracy (%) | Avg. Tokens |
|---|---|---|
| Random Run 1 | 86.8 | 1.0k |
| Random Run 2 | 87.2 | 1.2k |
| Random Run 3 | 86.6 | 1.1k |
| **Random Avg. (3 runs)** | 86.9 (±0.31) | 1.1k |
| *Main Reported (Ref.)* | *87.2* | *1.1k* |

**Implications for Debate Efficiency.** Recent theoretical works have debated the utility of multi-agent interaction. While some suggest debate inherently improves reasoning, others argue that without targeted intervention, debate processes can behave like random walks (Martingales) or collapse into groupthink. Our calibration results bridge this gap. By demonstrating that high uncertainty (low logprob / high entropy) accurately predicts potential errors, ours avoids the inefficiency of indiscriminate debate. Instead of forcing agents to debate on queries they have already mastered (High Logprob) or are completely ignorant of (Max Entropy), our framework leverages these calibrated proxies to selectively trigger debate only in the **"ambiguity zone"**, where peer review and expert escalation are most effective.

## 4.5. Robustness Analysis: Sensitivity to Agent Order

Our framework leverages supporting agents with varied prompts and sampling temperatures to incorporate heterogeneous perspectives. To investigate whether performance

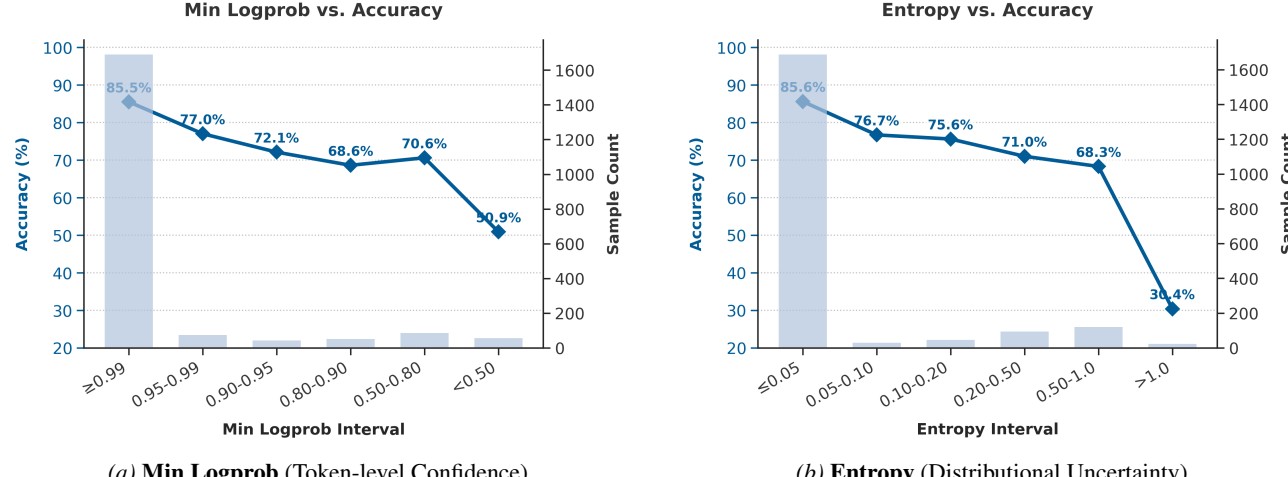

*(a)* **Min Logprob** (Token-level Confidence)  *(b)* **Entropy** (Distributional Uncertainty)

*Figure 3.* **Calibration Analysis of Confidence Proxies.** We analyze the relationship between accuracy and our two uncertainty metrics: (a) Min Logprob and (b) Entropy. The dual-axis charts show sample distribution (bars) and accuracy trend (lines). Crucially, accuracy consistently declines as uncertainty increases, **empirically validating our Monotone Calibration Assumption (Assumption 3.3)**. This confirms that instances flagged as uncertain are indeed prone to errors, establishing these metrics as effective gating signals for our adaptive framework.

relies on a specific interaction sequence, we compared our main setting against a scenario where the agent order is randomly permuted for every query.

As shown in Table 3, the randomized order yields an accuracy of 86.9%, which is statistically comparable to the main reported result of 87.2%, with a low standard deviation of 0.31. This minimal gap indicates that our method is effectively **permutation invariant**; the efficacy stems from the *presence* of diverse insights within the agent pool rather than a rigid curriculum. The stable token usage ($\sim$1.1k) further confirms that our adaptive gating mechanism remains robust and efficient even under stochastic interaction conditions.

## 5. Related Work

### 5.1. Multi-Agent Debate and Consensus Dynamics

Multi-agent debate (MAD) has emerged as a powerful paradigm to enhance LLM reasoning by overcoming the limitations of single-agent generation through iterative critique and refinement (Du et al., 2023; Liang et al., 2024; Chan et al., 2023). Early works demonstrated that multiple agents could correct hallucinations and improve factuality through unstructured dialogue (Du et al., 2023). Recent studies have further augmented this by assigning diverse personas or roles to agents to promote divergent thinking (Wang et al., 2024b).

However, theoretical analyses provide conflicting views on the efficacy of such interactions. Hu et al. (2025) formulate debate as Bayesian inference, proving that with a correct initial seed, debate converges to the truth. In contrast, Choi

et al. (2025) argue that under symmetric conditions—where information flows equally among homogeneous agents—the belief update process forms a martingale, yielding no expected improvement. Our work reconciles these perspectives. We explicitly design LASE to break the symmetry condition identified by Choi et al. (2025). By adopting a directed star topology (Leader-Supporter) rather than an all-to-all mesh, LASE transforms neutral information mixing into a directed optimization toward correctness.

### 5.2. Efficient and Adaptive Reasoning

While effective, standard debate frameworks incur significant computational costs due to indiscriminate agent activation (Liu et al., 2024a). To mitigate this, Li et al. (2024) proposed sparse communication topologies; however, their approach relies on static graph structures. Other approaches have explored optimizing collaboration through Reinforcement Learning (RL) (Liu et al., 2024b; Estornell et al., 2024), but these introduce substantial training overhead.

In contrast, ours focuses on *test-time* efficiency without parameter updates. Unlike self-correction methods that rely solely on a single agent's internal capability (Shinn et al., 2023), ours leverages intrinsic uncertainty signals (e.g., token-level log-probabilities) to dynamically allocate computation. By selectively escalating from single-agent reasoning to multi-agent collaboration only when necessary ($U = 1$), LASE achieves a Pareto-optimal balance between accuracy and token cost, outperforming both static ensembles (Wang et al., 2022) and fixed-topology debates.

## 6. Conclusion

In this work, we proposed LASE, a cost-efficient framework for adaptive multi-agent reasoning. Motivated by the inefficiency of indiscriminate debate, we introduced a confidence-guided mechanism that dynamically allocates computational resources only when necessary. By adopting a leader-supporter topology to break symmetry, LASE overcomes the theoretical limitations of martingale neutrality, inducing a directed corrective bias toward the ground truth.

## Impact Statement

This paper presents work whose goal is to advance the field of Machine Learning, specifically in efficient and reliable multi-agent reasoning. Our proposed framework, LASE, significantly reduces the computational cost (token consumption) associated with complex reasoning tasks, contributing to the broader goal of Green AI by lowering the energy footprint of large-scale model deployment. Furthermore, by integrating uncertainty gating and adversarial verification, our approach aims to mitigate hallucinations and improve the reliability of automated decision-making systems. While general risks associated with Large Language Models (e.g., potential for misuse or bias) remain, our work primarily focuses on enhancing the efficiency and trustworthiness of these systems without introducing specific new negative societal consequences.

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

# A. Proofs for Theoretical Claims

**Setup and events.** Let $L$ denote the leader and $\mathcal{P}$ the set of peers. Let $U := U(z_L^{(0)}) \in \{0, 1\}$ be the uncertainty event from Definition 3.1. Write the leader-only extracted answer as $\hat{y}_{\text{solo}} := e(z_L^{(0)})$ and the LASE output as $\hat{y}_{\text{LASE}}$. Let $S$ denote the event that, when LASE queries a peer on $U = 1$, the peer response set contains at least one *correct seed* in the sense of Assumption 2.3.

## A.1. Auxiliary assumptions for the gain bound

For a clean, rigorous lower bound on selective interaction, we use the following mild assumptions.

**Assumption A.1** (Seed prevalence on uncertain instances). There exists $\rho \in (0, 1]$ such that

$$\mathbb{P}(S \mid U = 1) \geq \rho. \tag{21}$$

**Assumption A.2** (Conditional amplification given a seed). There exists $\delta > 0$ such that

$$\mathbb{P}(\hat{y}_{\text{LASE}} = y \mid U = 1, S) \geq \mathbb{P}(\hat{y}_{\text{solo}} = y \mid U = 1) + \delta. \tag{22}$$

**Assumption A.3** (Conservativeness when no seed). When no correct seed is observed, LASE is (in expectation) no worse than keeping the leader-only answer:

$$\mathbb{P}(\hat{y}_{\text{LASE}} = y \mid U = 1, S^c) \geq \mathbb{P}(\hat{y}_{\text{solo}} = y \mid U = 1). \tag{23}$$

**Remark.** Assumption A.3 is satisfied, for example, if the leader is allowed to retain $\hat{y}_{\text{solo}}$ whenever the interaction does not provide a compelling correction. Assumption A.2 abstracts the "corrective regime" effect captured by the latent-concept Bayes view (Theorem 2.4).

## A.2. Proof of Proposition: breaking symmetry

**Proposition A.4** (LASE breaks symmetry conditions underlying martingale neutrality). *Consider a discrete-choice abstraction as in Choi et al. (2025), with posterior-mean beliefs $p_{i,t}$ updated by neighbor information. Under LASE's leader-centric star topology (only $L$ incorporates peer messages) together with uncertainty-gated interaction (an update is executed only when $U = 1$, where $U$ is measurable w.r.t. the leader transcript), the sufficient symmetry/averaging condition required by Theorem 2.7 is not enforced for the leader update. Therefore, Theorem 2.7 cannot be invoked to conclude martingale neutrality for LASE.*

*Proof.* Theorem 2.7 states a *sufficient* condition: if, at an update step, an agent's belief equals the average of its neighbors' beliefs, then the agent's next-step belief is a martingale in expectation. Formally, let $\mathcal{N}_i$ denote the neighbor set whose messages agent $i$ incorporates at that step; the sufficient condition is

$$p_{i,t-1} = \frac{1}{|\mathcal{N}_i|} \sum_{j \in \mathcal{N}_i} p_{j,t-1}. \tag{24}$$

Under LASE, the interaction graph is a directed star: only the leader incorporates peer information, so on any step where interaction occurs, $\mathcal{N}_L \subseteq \mathcal{P}$ (often $|\mathcal{N}_L| = 1$), while peers do not incorporate other peers' messages in their own updates. Thus the homogeneous fully-connected exchangeable setting used to motivate (24) for all agents does not apply.

Moreover, interaction is gated: the leader update is executed only on the event $U = 1$, where $U$ is measurable w.r.t. the leader's transcript. Condition (24) for the leader would then require that, on every executed update,

$$p_{L,t-1} = \frac{1}{|\mathcal{N}_L|} \sum_{j \in \mathcal{N}_L} p_{j,t-1} \quad \text{(on the event that the update is executed).} \tag{25}$$

LASE does not impose (25) as a design constraint; it only specifies *when* and *from whom* the leader receives messages (via $U$ and $\mathcal{N}_L$). Hence the sufficient symmetry condition of Theorem 2.7 is not guaranteed for the leader updates under LASE. Therefore, the martingale neutrality conclusion of Theorem 2.7 cannot be applied to characterize LASE's expected belief drift. $\square$

## A.3. Proof of Theorem: selective gain

**Theorem A.5** (Selective debate yields positive expected improvement). *Under Assumptions A.1–A.3,*

$$\text{Acc}(\pi_{LASE}) \geq \text{Acc}(\pi_{solo}) + \mathbb{P}(U = 1) \cdot \rho\delta. \tag{26}$$

*Proof.* By definition, LASE outputs $\hat{y}_{\text{solo}}$ when $U = 0$, and outputs $\hat{y}_{\text{LASE}}$ (after interaction) when $U = 1$. Therefore,

$$\begin{aligned}
\text{Acc}(\pi_{\text{LASE}}) &= \mathbb{P}(\hat{y}_{\text{solo}} = y, \, U = 0) + \mathbb{P}(\hat{y}_{\text{LASE}} = y, \, U = 1) \\
&= \mathbb{P}(U = 0)\mathbb{P}(\hat{y}_{\text{solo}} = y \mid U = 0) + \mathbb{P}(U = 1)\mathbb{P}(\hat{y}_{\text{LASE}} = y \mid U = 1).
\end{aligned} \tag{27}$$

Similarly,

$$\text{Acc}(\pi_{\text{solo}}) = \mathbb{P}(\hat{y}_{\text{solo}} = y) = \mathbb{P}(U = 0)\mathbb{P}(\hat{y}_{\text{solo}} = y \mid U = 0) + \mathbb{P}(U = 1)\mathbb{P}(\hat{y}_{\text{solo}} = y \mid U = 1). \tag{28}$$

Subtracting (28) from (27) yields

$$\text{Acc}(\pi_{\text{LASE}}) - \text{Acc}(\pi_{\text{solo}}) = \mathbb{P}(U = 1)\Big(\mathbb{P}(\hat{y}_{\text{LASE}} = y \mid U = 1) - \mathbb{P}(\hat{y}_{\text{solo}} = y \mid U = 1)\Big). \tag{29}$$

Now condition on the seed event $S$:

$$\begin{aligned}
\mathbb{P}(\hat{y}_{\text{LASE}} = y \mid U = 1) &= \mathbb{P}(\hat{y}_{\text{LASE}} = y \mid U = 1, S)\mathbb{P}(S \mid U = 1) \\
&\quad + \mathbb{P}(\hat{y}_{\text{LASE}} = y \mid U = 1, S^c)\mathbb{P}(S^c \mid U = 1) \\
&\geq \Big(\mathbb{P}(\hat{y}_{\text{solo}} = y \mid U = 1) + \delta\Big)\mathbb{P}(S \mid U = 1) + \mathbb{P}(\hat{y}_{\text{solo}} = y \mid U = 1)\mathbb{P}(S^c \mid U = 1) \\
&= \mathbb{P}(\hat{y}_{\text{solo}} = y \mid U = 1) + \delta\,\mathbb{P}(S \mid U = 1),
\end{aligned} \tag{30}$$

where the inequality uses Assumptions A.2 and A.3. By Assumption A.1, $\mathbb{P}(S \mid U = 1) \geq \rho$, so

$$\mathbb{P}(\hat{y}_{\text{LASE}} = y \mid U = 1) \geq \mathbb{P}(\hat{y}_{\text{solo}} = y \mid U = 1) + \rho\delta. \tag{31}$$

Plugging this into (29) yields

$$\text{Acc}(\pi_{\text{LASE}}) - \text{Acc}(\pi_{\text{solo}}) \geq \mathbb{P}(U = 1) \cdot \rho\delta, \tag{32}$$

completing the proof. $\square$

## A.4. Proof of Proposition: interaction-topology cost scaling

**Proposition A.6** (Interaction-topology cost scaling). *Consider $T$ rounds and $n$ agents. Define an* ordered cross-agent context inclusion *to be an ordered pair $(i, j)$ with $i \neq j$ such that agent $i$ conditions its generation at some round on the most recent message produced by agent $j$. In an all-to-all debate where each agent conditions on every other agent's latest message each round, the number of ordered cross-agent context inclusions scales as $\Theta(Tn^2)$. In a leader-centric scheme that queries at most $k = O(1)$ peers per round and only the leader incorporates peer messages, the number of ordered cross-agent context inclusions is $\mathcal{O}(Tk) = \mathcal{O}(T)$, independent of $n$.*

*Proof.* **All-to-all.** At each round $t \in \{1, \ldots, T\}$, each agent $i$ conditions on the latest messages from all other agents $j \neq i$. Thus the number of ordered inclusions per round is $n(n - 1)$, and over $T$ rounds it is

$$T \cdot n(n - 1) = \Theta(Tn^2). \tag{33}$$

**Leader-centric with at most $k$ peers.** At each round, LASE queries at most $k$ peers. Each queried peer conditions on the leader's message (one inclusion per peer), and the leader conditions on each queried peer's message (one inclusion per peer). Hence there are at most $2k$ ordered inclusions per round, and over $T$ rounds at most $2kT = \mathcal{O}(Tk) = \mathcal{O}(T)$, independent of $n$. $\square$

## A.5. Proof of Corollary: Pareto frontier improvement

**Corollary A.7** (Pareto frontier improvement). *Assume* $\mathbb{P}(U = 1) \in (0, 1)$ *and Assumptions A.1–A.3. Let* $\pi_{solo}$ *be leader-only inference and let* $\pi_{full}$ *be an* always-on *variant of LASE that executes the same interaction pipeline that LASE would execute on* $U = 1$, *but does so for every instance (i.e., forces interaction regardless of* $U$). *Then* $\mathrm{Cost}(\pi_{LASE}) < \mathrm{Cost}(\pi_{full})$ *and* $\mathrm{Acc}(\pi_{LASE}) > \mathrm{Acc}(\pi_{solo})$. *Moreover, for any budget* $B$ *with* $\mathrm{Cost}(\pi_{LASE}) \leq B < \mathrm{Cost}(\pi_{full})$, $\pi_{full}$ *is infeasible under* $B$ *while* $\pi_{LASE}$ *is feasible and satisfies*

$$\mathrm{Acc}(\pi_{LASE}) \geq \mathrm{Acc}(\pi_{solo}) + \mathbb{P}(U = 1)\rho\delta. \tag{34}$$

*Proof.* **Cost comparison.** Let $C_0$ denote the expected cost of the leader-only stage (incurred by both policies), and let $C_+ > 0$ denote the expected additional interaction cost of executing the LASE interaction pipeline (peer messages plus leader updates), matching the pipeline used by $\pi_{\mathrm{full}}$. By construction, $\pi_{\mathrm{full}}$ always executes the interaction pipeline, so

$$\mathrm{Cost}(\pi_{\mathrm{full}}) = C_0 + C_+. \tag{35}$$

LASE executes interaction iff $U = 1$, so

$$\mathrm{Cost}(\pi_{\mathrm{LASE}}) = C_0 + \mathbb{P}(U = 1)\, C_+. \tag{36}$$

Since $\mathbb{P}(U = 1) \in (0, 1)$ and $C_+ > 0$, we obtain $\mathrm{Cost}(\pi_{\mathrm{LASE}}) < \mathrm{Cost}(\pi_{\mathrm{full}})$.

**Accuracy comparison.** By Theorem A.5,

$$\mathrm{Acc}(\pi_{\mathrm{LASE}}) \geq \mathrm{Acc}(\pi_{\mathrm{solo}}) + \mathbb{P}(U = 1)\rho\delta. \tag{37}$$

The improvement is strict because $\mathbb{P}(U = 1) > 0$, $\rho > 0$, and $\delta > 0$.

**Budget separation.** Fix any $B$ such that $\mathrm{Cost}(\pi_{\mathrm{LASE}}) \leq B < \mathrm{Cost}(\pi_{\mathrm{full}})$. Then $\pi_{\mathrm{full}}$ is infeasible under $B$ by definition, while $\pi_{\mathrm{LASE}}$ is feasible and attains the stated accuracy lower bound. This establishes that LASE provides a strictly better feasible accuracy–cost point than always-on interaction at such budgets. $\square$

# B. Extended Theoretical Discussion and Connection to Practice

In Appendix A, we established the theoretical foundation demonstrating that LASE breaks martingale neutrality (Proposition A.4) and yields positive expected improvement under specific assumptions (Theorem A.5). In this section, we bridge these formal guarantees with the empirical dynamics observed in our experiments, addressing how theoretical abstractions translate into practical multi-agent interactions.

## B.1. Breaking Martingale Neutrality in Practice

Theoretical models of symmetric debate often assume that agents update their beliefs by uniformly averaging neighboring signals (Choi et al., 2025), leading to a zero-drift martingale process. While Proposition A.4 formally shows that LASE's topology invalidates this symmetry condition, the practical implication is profound. Collaboration alone does not guarantee reasoning improvement; unstructured debate can often lead to stagnation or the reinforcement of incorrect reasoning (echo chambers). LASE transforms this neutral information-mixing into a *corrective update process*. By adopting a leader-centric design, the leader maintains strict control over the reasoning trajectory. Supporters do not overwrite the leader's belief symmetrically; instead, they provide targeted critiques that the leader aggregates, enforcing a directed and biased update toward error correction.

## B.2. The Practical Role of Theoretical Assumptions: Corrective Pressure

Theorem A.5 relies on the existence of a "correct seed" (Assumption 3.4) and subsequent signal amplification (Assumption 3.5). While a perfectly correct seed is an idealization, LASE operationalizes this concept robustly in practice.

The framework does not strictly require a flawless seed from a supporter to improve correctness. Instead, it induces a functionally equivalent *corrective pressure* through structured adversarial feedback (e.g., the "Devil's Advocate" role detailed in Appendix D). When the leader's confidence is low, this adversarial check actively challenges intermediate reasoning steps and exposes logical traps. Rather than relying on a perfect initial seed to amplify, LASE relies on this targeted feedback to force the leader to revise flawed trajectories. Consequently, the theoretical assumption of seed amplification manifests practically as the mitigation of error propagation under uncertainty.

### B.3. Empirical Validation on the Uncertain Subset

A core premise of LASE is that peer interaction is selectively beneficial only in uncertain regimes (where $U = 1$). This is empirically validated by isolating the source of our accuracy gains. By design, when the uncertainty event does not occur ($U = 0$), LASE completely bypasses interaction and defaults exactly to the single-agent output.

Therefore, the performance improvements observed across our benchmarks (e.g., an average gain of $+4.1\%$p across base models on MATH-500 over the single-agent baseline) are derived *entirely* from the uncertain subset ($U = 1$). This direct empirical evidence confirms the theoretical motivation: peer interaction, when unstructured, may be neutral or detrimental, but when explicitly gated by confidence proxies (Definition 3.1), it perfectly isolates the regime where corrective interaction is strictly beneficial.

## C. Datasets Details

Our evaluation employs six benchmarks across mathematical reasoning and general knowledge. For Mathematical Reasoning, we utilize two subsets derived from the MATH benchmark (Hendrycks et al., 2021). MATH-500 (Lightman et al., 2023) is a curated test set of 500 problems widely used for evaluating rigorous reasoning steps. MATH-lv5 is a custom subset we constructed by filtering only "Level 5" (highest difficulty) problems, resulting in 1,324 samples. Additionally, we use GSM-Symbolic (Mirzadeh et al., 2024), a dataset designed by Apple to test genuine reasoning capabilities against rote memorization. It generates symbolic variations (altering names, numbers, and contexts) from original GSM8K problems. From the original pool of 5,000 samples (100 base templates $\times$ 50 variations), we sampled 5 variations per template, totaling 500 samples. For Knowledge & Safety, we use MMLU-Pro (Wang et al., 2024a) ($N = 504$, stratified sampling) and GPQA-Diamond (Rein et al., 2024) ($N = 198$) to assess expert-level knowledge. We also employ TruthfulQA-MC (Lin et al., 2022) ($N = 817$) to evaluate the model's robustness against hallucinations.

## D. Multi-Agent System Prompts

This section details the system prompts used for each agent in our multi-agent reasoning framework. All agents are instructed to follow specific roles to enhance logical rigor and problem-solving accuracy.

| 1. Leading Agent |
| --- |
| **System Prompt:** |
| You are a lead problem solver. Solve the problem step by step with careful reasoning. Put your final answer in \boxed{}. |
| **User Prompt:** |
| Problem: {problem} |
| Provide your solution step by step. |
| IMPORTANT: Put your final numerical answer inside \boxed{}. The answer must be a plain number only - NO units, NO currency symbols, NO thousands separators, and NO percentage signs. Just the raw number. |

| 2. Supporting Agent 1 |
| --- |
| **System Prompt:** |
| You are a helpful collaborator. Help the problem solver by providing hints, alternative perspectives, or clarifications. |
| IMPORTANT RULES: |
| 1. Do NOT solve the problem directly - guide them to the right approach. |
| 2. Do NOT use \boxed{} in your response - that is reserved for the lead solver. |
| 3. Focus on explaining concepts, pointing out where they went wrong, or suggesting strategies. |
| **User Prompt:** |
| Problem: {problem} |
| Current attempt: {post_adversarial_response} |
| A reviewer raised these concerns: {adversarial_response} |
| The solver seems uncertain. Provide helpful hints or perspectives to guide them toward the correct solution. Remember: Do NOT give the final answer or use \boxed{}. |

---

**3. Supporting Agent 2**

**System Prompt:**
You are a senior expert consultant. A team is struggling with a difficult problem. Your role is to provide expert guidance and insights, NOT to solve the problem for them.
IMPORTANT RULES:
1. Help them understand the key concepts they're missing.
2. Point out flaws in their reasoning and suggest the right direction.
3. Do NOT provide the final answer directly.
4. Do NOT use \boxed{} in your response - that is reserved for the lead solver.
**User Prompt:**
Problem: {problem}
Current solution attempt (after receiving support): {post_support_response}
Reviewer concerns: {adversarial_response}
Collaborator's hints: {supporting_response}
The solver is still uncertain even after receiving hints. As the expert, identify the critical insight or concept they're missing and guide them toward the correct approach. Explain WHY their current reasoning may be flawed. Remember: Do NOT give the final answer or use \boxed{}. Let them solve it.

---

**4. Supporting Agent 3**

**System Prompt:**
You are an aggressive Devil's Advocate. Your job is to ATTACK the solution, not to validate it.
YOUR MISSION:
1. FIND THE WEAK POINT: Every solution has a vulnerability. Find it.
2. SET LOGICAL TRAPS: Ask questions that expose hidden assumptions or flawed reasoning.
3. DEMAND RIGOROUS PROOF: 'How do you KNOW this step is valid?' 'Prove it.'
4. CHALLENGE EDGE CASES: 'What if $x = 0$? What if the denominator is zero?'
DO NOT:
- Simply say 'looks correct' or 'well done'
- Make up fake mathematical errors
- Be satisfied with hand-wavy explanations
- Use \boxed{} - that is reserved for the lead solver
Your goal is to make the solver DEFEND their work rigorously. If they can't defend it convincingly, their solution is weak.
**User Prompt:**
Problem: {problem}
Proposed solution: {leader_response}
ATTACK this solution. Find the logical trap or weak point. Ask pointed questions that force the solver to justify every step. Do NOT accept the solution at face value. Remember: Do NOT provide your own answer or use \boxed{}.

## E. Scalability to Open-Weight Models: Llama-3-8B

The primary empirical evaluation in Section 4 focused on frontier proprietary models (Gemini-2.0-Flash and GPT-4.1-mini). To demonstrate the generalization of our framework beyond specific model families and to investigate its behavior in low-capability regimes, we conducted an additional evaluation using a smaller, open-weight model (Llama-3-8B) on the MATH-500 benchmark.

### E.1. Performance and Efficiency

As shown in Table 4, LASE consistently outperforms the single-agent baseline, Majority Voting (MV), and Multi-Agent Debate (MAD). While achieving the highest accuracy of 32.9%, LASE requires significantly fewer tokens (1.4M) compared to MV (2.0M) and MAD (4.4M). This confirms that our confidence-guided, asymmetric framework extends effectively to open-weight models.

*Table 4.* Performance comparison on MATH-500 using Llama-3-8B.

| Method | Accuracy | Total Tokens |
|---|---|---|
| Single Agent | 25.7% | 0.3M |
| Majority Voting (MV) | 31.8% | 2.0M |
| Multi-Agent Debate (MAD) | 24.1% | 4.4M |
| **Ours (LASE)** | **32.9%** | **1.4M** |

### E.2. Adaptive Triggering and Symmetric Debate Collapse

When the base model is weaker, confidence scores naturally drop, triggering interaction more frequently. Consequently, LASE's token usage scales dynamically based on the model's capability (consuming roughly 4.6x the tokens of a single agent for Llama-3-8B, compared to 1.8x for GPT-4.1-mini). This demonstrates that our gating policy is not dormant but adaptively reallocates compute budget to compensate for higher uncertainty.

Furthermore, we observe that unstructured symmetric debate (MAD) actually degrades performance (24.1%) compared to the single-agent baseline (25.7%). In low-capability regimes, models often lack a correct seed (Assumption 3.4), causing symmetric interactions to blindly amplify hallucinations and errors. LASE's structured, leader-centric design explicitly prevents this symmetric collapse by isolating and carefully validating peer feedback.

## F. Sensitivity Analysis and Pareto Frontier

To ensure the robustness of our confidence-gating mechanism and verify that performance does not rely on highly specific hyperparameter configurations, we conducted a comprehensive threshold sweep for token log-probability ($\tau_{lp}$) and entropy ($\tau_h$) on MATH-500 using GPT-4.1-mini.

**Robustness of the Accuracy-Cost Tradeoff.** Table 5 presents the results of varying $\tau_{lp}$ from 0.50 to 0.95 while keeping $\tau_h$ fixed at 0.50. The accuracy remains remarkably stable, operating within a tight range of 74.1% to 75.1%.

Instead of drastically altering correctness, adjusting the threshold primarily scales the trigger rate (from 21.4% to 42.8%) and, consequently, the token usage. This confirms that our thresholds act as reliable control knobs along a stable accuracy-cost Pareto frontier, allowing practitioners to explicitly trade off computational budget for marginal accuracy gains without destabilizing the reasoning process.

**Stability Across Uncertainty Metrics.** Furthermore, this robust stability extends to the distributional uncertainty proxy as well. When sweeping the entropy threshold ($\tau_h$) across different values while keeping $\tau_{lp}$ fixed, we observed a similarly tight accuracy range. For instance, testing $\tau_h \in \{0.2, 0.5, 0.8\}$ yielded accuracies of 73.8%, 74.2%, and 74.1%, respectively (a narrow range with a maximum gap of $\Delta = 0.4\%$p). This confirms that neither confidence metric relies on cherry-picked heuristics, and both consistently gate interaction in a controlled and predictable manner.

*Table 5.* Threshold sweep analysis on MATH-500 (GPT-4.1-mini).

| $\tau_{lp}$ | $\tau_h$ | **Accuracy** | **Avg. Tokens** | **Esc Rate** ($U = 1$) |
|---|---|---|---|---|
| 0.50 | 0.50 | 74.1% | 1.8k | 21.4% |
| 0.60 | 0.50 | 74.8% | 1.8k | 21.2% |
| 0.80 | 0.50 | 74.2% | 1.8k | 22.6% |
| 0.90 | 0.50 | 74.8% | 2.6k | 30.8% |
| 0.95 | 0.50 | 75.1% | 3.1k | 42.8% |

## G. Comparison with Additional Multi-Agent Frameworks

To properly situate LASE within the broader landscape of LLM-agent reasoning, we provide an extended empirical comparison with recent multi-agent topologies and contextualize our approach within neighboring reasoning paradigms.

### G.1. Empirical Comparison with Mixture-of-Agents (MoA)

Recent frameworks such as Mixture-of-Agents (MoA) (Wang et al., 2025) achieve strong performance by routing inputs through multiple layers of diverse LLMs. However, these systems typically employ a static, always-on routing strategy, invoking the full multi-agent pipeline for every query regardless of its difficulty.

To evaluate the efficiency of our adaptive approach against such dense topologies, we conducted an additional comparison with MoA on the MATH-500 benchmark using GPT-4.1-mini. As shown in Table 6, LASE achieves an accuracy of 74.2%, outperforming MoA (72.0%) by +2.2%p. Crucially, LASE accomplishes this while consuming only ~25% of the computational budget (0.9M vs. 3.6M tokens).

This significant efficiency gain highlights the advantage of confidence-guided interaction. Because LASE is not a fixed topology but an adaptive policy, it completely bypasses multi-agent interaction on highly confident queries, focusing compute exclusively on uncertain regimes. In this regard, LASE is entirely complementary to fixed-routing frameworks like MoA and could potentially be integrated as a higher-level gating mechanism to optimize their routing costs.

*Table 6.* Performance and token cost comparison with Mixture-of-Agents (MoA) on MATH-500 using GPT-4.1-mini.

| Method | Accuracy | Total Tokens |
|---|---|---|
| Single Agent | 73.0% | 0.5M |
| Mixture-of-Agents (MoA) | 72.0% | 3.6M |
| **Ours (LASE)** | **74.2%** | **0.9M** |

### G.2. Positioning within Broader Reasoning Paradigms

Beyond standard debate, LASE connects deeply with several prominent branches of LLM reasoning:

**Self-Correction vs. External Feedback.**    Self-correction frameworks (e.g., Reflexion; Shinn et al., 2023) attempt to refine outputs through internal critique. However, models often struggle to identify their own logical flaws, leading to degenerate self-correction loops where correct answers are sometimes revised into incorrect ones. LASE mitigates this by introducing structured *external* feedback through diverse supporter roles, injecting targeted corrective pressure that a single agent cannot generate independently.

**Unifying Adaptive Routing and Verifier/Critic Systems.**    Ultimately, LASE can be viewed as an integrated framework that unifies two distinct paradigms. It employs *adaptive routing* (via token-level uncertainty gating) to decide *when* to allocate additional compute, while leveraging a *verifier/critic-style interaction* (via the Leader-Supporter topology) to dictate *how* that compute should be structured. This combination prevents the cascading hallucination errors often seen in unstructured multi-agent setups, resulting in a highly reliable and economically scalable reasoning system.

## H. Qualitative Analysis and Failure Modes

To better understand the boundaries of the LASE framework and address its limitations, we provide a qualitative analysis of its primary failure modes. While our confidence-guided interaction effectively mitigates many reasoning errors, the remaining failures generally fall into two distinct categories: miscalibrated overconfidence and execution failure post-critique.

**Confident Errors (Overconfidence).**    The first failure mode occurs when the leader model produces an incorrect answer but exhibits extremely high internal confidence (e.g., $\text{LP}_{\min} \geq 0.99$). In these instances, the uncertainty event does not trigger ($U = 0$), and LASE intentionally bypasses the multi-agent interaction phase. As shown in our calibration analysis (Figure 3a), a small fraction of highly confident generations are indeed incorrect.

While it might seem intuitive to force interaction on all queries to catch these errors, empirical observations reveal two critical bottlenecks. First, when a model is highly confident in a flawed reasoning path, it is notoriously resistant to external feedback. Second, because our current instantiation utilizes homogeneous agents, the supporters often share the same intrinsic capacity limits and blind spots as the leader. Consequently, forcing debate in these scenarios frequently results in an echo chamber where supporters merely validate the leader's hallucination, yielding diminishing returns relative to the token cost incurred. Therefore, leaving these confident errors uncorrected is not a flaw of the gating mechanism, but rather a deliberate accuracy-efficiency tradeoff designed to focus computational resources solely on the ambiguity zone where marginal gains are highest.

**Execution Failure Post-Critique.**    The second failure mode occurs in the uncertain regime ($U = 1$) where the gating mechanism successfully triggers interaction. In these cases, the supporting agent (e.g., the Devil's Advocate) accurately identifies a critical logical flaw or edge case, and the leader explicitly acknowledges this vulnerability.

However, despite recognizing the error, the leader ultimately fails to arrive at the correct final answer. This typically happens because the multi-step algebraic manipulation or logical revision required to fix the error exceeds the base model's intrinsic

reasoning capacity. In other words, while the external corrective pressure successfully breaks the flawed trajectory, the base model lacks the execution capability to reconstruct the correct path. This highlights a fundamental limitation of test-time interaction frameworks: their performance ceiling is ultimately bounded by the raw reasoning and execution capabilities of the underlying base model.

## I. Limitations and Future Work

In this work, we implemented the LASE framework adhering to the **theoretical minimal requirements** necessary to break the martingale symmetry and induce corrective updates. To maximize token efficiency, our current topology restricts interaction to a leader-centric 1-on-1 dynamic, activating a single supporter at a time. While this successfully demonstrates the efficacy of adaptive role differentiation, it may not fully exploit the collective intelligence available in larger ensembles.

A promising direction for future work is to generalize the supporter role. Instead of a single-agent interaction, the support mechanism could be replaced by a **multi-agent debate subsystem** or a small ensemble. For instance, the leader could query a group of supporters who debate among themselves before synthesizing a consensus signal. While this would increase computational cost, such a hierarchical expansion could provide more robust auxiliary signals ($\Delta_t$) for extremely complex tasks, further pushing the Pareto frontier of accuracy and efficiency.

