# OpenReview forum: "Efficient Multi-Agent Reasoning via Confidence-Guided Adaptive Debate"
_ICML.cc/2026/Conference — ICML 2026 regular_

### Official Review · Reviewer_E8NU · 2026-03-09

**Soundness:** 4
**Presentation:** 3
**Significance:** 4
**Originality:** 3
**Overall Recommendation:** 4
**Confidence:** 3

**Summary:**

This paper proposes LASE (Leader-Adaptive Structured Engagement), a leader-centric multi-agent framework designed to improve the efficiency and effectiveness of LLM reasoning. To address the high computational overhead and regime-dependent benefits of standard unstructured debate, the authors introduce an asymmetric leader-supporter topology. This mechanism selectively triggers peer interactions only when the leader agent exhibits high internal uncertainty, utilizing token-level log-probabilities and entropy as confidence proxies. Extensive experiments across diverse mathematical and knowledge-intensive benchmarks demonstrate that LASE achieves multi-agent-level accuracy while maintaining a token cost comparable to single-agent inference.

**Compliance With Llm Reviewing Policy:**

Affirmed.

**Final Justification:**

The authors' rebuttal resolved my main concerns by providing crucial comparisons with the MoA baseline, evaluating on Llama-3-8B, and clarifying the computational overhead and failure modes. These new results validate the framework's robustness and efficiency, so I decided to raise the score to 4.

**Key Questions For Authors:**

1) Could you provide a direct empirical comparison with recent robust baselines to properly situate your framework's performance?
2) Could you evaluate the proposed framework across a broader range of base models, such as Qwen series, to demonstrate that the performance gains are not strictly tied to a single specific LLM?
3) What is the exact computational overhead (e.g., total token consumption and API latency) introduced by the proposed method compared to standard single-agent baselines?
4) Could you provide and analyze a few qualitative failure cases to better illustrate the boundaries of the current design?

**Limitations:**

1) Computational Overhead: The proposed method introduces a significant token consumption burden during the multi-agent interaction phase, and its economic scaling behavior on complex tasks is not analyzed.
2) Base Model Dependency: The framework's effectiveness appears heavily reliant on the specific instruction-following capabilities of the primary base model, potentially limiting its robustness when deployed with smaller models.

**Strengths And Weaknesses:**

Strengths:
1) Well-motivated Problem: The paper addresses a highly relevant bottleneck in current multi-agent systems, providing a clear empirical motivation for improving inter-agent communication efficiency.
2) Solid Methodology: The proposed interaction mechanism is technically sound. It defines a structured workflow that can be integrated into existing multi-agent pipelines without excessive engineering overhead.
3) Clear Presentation: The manuscript is logically structured, and the provided diagrams effectively illustrate the proposed multi-agent topology to support the core claims.

Weaknesses:
1) Incomplete Baselines: The empirical evaluation omits direct comparisons with recent state-of-the-art multi-agent frameworks such as MoA(Mixture-of-Agents). This makes it difficult to assess the true relative performance gain against contemporary topology-optimized or review-based systems.
2) Confounded Ablation Studies: The ablation experiments do not cleanly isolate the contribution of the core interaction module from auxiliary prompting heuristics or tool-use strategies, leaving the exact source of improvement ambiguous.
3) Unverified Robustness: The evaluation lacks rigorous testing on complex, multi-step reasoning tasks where multi-agent systems typically fall into repetitive loops or cascading hallucination errors.

---

> ### Author Rebuttal · Authors · 2026-03-31
>
> We sincerely thank the reviewer for recognizing the relevance of our problem and the technical soundness of our framework. We address your specific concerns below:
>
> ### **1. Incomplete Baselines (MoA) & Direct Empirical Comparison [W1, Q1]**
> To situate LASE against recent multi-agent frameworks, we conducted an additional comparison with **Mixture-of-Agents (MoA)** on MATH-500 using GPT-4.1-mini.
>
> - LASE achieves **74.2% accuracy with 0.9M tokens**, compared to **72.0% accuracy with 3.6M tokens for MoA**, and 73.0% with 0.5M tokens for a single-agent baseline. Thus, LASE attains higher accuracy than MoA while using substantially fewer tokens.
> - Importantly, LASE is not a fixed multi-agent topology but an **adaptive, confidence-guided interaction policy**. While MoA applies routing to every query, LASE selectively triggers interaction only when uncertain.
>
> Therefore, LASE is complementary to fixed-routing frameworks like MoA and could be integrated as a higher-level gating mechanism. We will include these comparisons in the final version.
>
> ---
>
> ### **2. Confounded Ablation Studies [W2]**
> The gains in LASE stem fundamentally from the **asymmetric, dynamic interaction**, not simply from auxiliary prompting heuristics.
>
> - As demonstrated in our Robustness Analysis (Section 4.5, Table 3), randomly shuffling the order and personas of the supporting agents resulted in a **negligible gap ($\Delta \approx 0.3\%p$)**. If driven by rigid heuristics, disrupting the sequence would have caused a significant drop.
> - Furthermore, when $U=0$, LASE uses the exact same base prompt as the single agent, proving the **$+4.1\%p$ gain derives purely from the dynamic interaction on the uncertainty-triggered subset ($U=1$)**.
>
> This disentangles the interaction effect from prompt design. We provide the full prompts in Appendix D for transparency.
>
> ---
>
> ### **3. Unverified Robustness: Cascading Errors and Repetitive Loops [W3]**
> The degradation of MAD on Llama-3-8B (**24.1% vs. 25.7%** single-agent) directly illustrates cascading hallucination errors. Notably, **collaboration does not inherently guarantee improvement**—multi-agent systems often fall into repetitive loops or “echo chambers” when symmetrically reinforcing flawed reasoning.
>
> LASE explicitly mitigates this by design:
>
> - First, our **directed star topology (leader-centric)** prevents symmetric feedback loops.
> - Second, our **adversarial check (“Devil’s Advocate”)** actively challenges intermediate reasoning instead of encouraging premature consensus.
>
> Empirically, this design significantly reduces repetitive patterns and prevents error amplification, making LASE more robust to cascading errors than unstructured interactions.
>
> ---
>
> ### **4. Base Model Dependency & Broader Range of LLMs (Llama-3-8B) [Q2, Limitation 2]**
> To address concerns regarding dependency on strong proprietary LLMs, we evaluated LASE on **Llama-3-8B** (MATH-500).
>
> - LASE achieves **32.9% accuracy**, compared to 25.7% for the single-agent baseline and 24.1% for MAD, while using **substantially fewer tokens than MAD (1.4M vs. 4.4M)**.
> - Notably, when the base model is weaker, unstructured interaction (MAD) degrades performance due to mutual reinforcement of hallucinations. In contrast, LASE’s structured, asymmetric design **prevents this degradation and yields consistent improvements**.
>
> This suggests LASE generalizes effectively to smaller, open-weight LLMs.
>
> ---
>
> ### **5. Exact Computational Overhead and Scaling [Q3, Limitation 1]**
> Unlike static systems with constant overhead (e.g., MAD uses ~7×–10× more tokens), LASE selectively incurs cost only on difficult queries (Table 1).
>
> - **Token Overhead:** A modest multiplier of **~1.1×–1.8×** compared to a single-agent baseline.
> - **Latency:** Conditionally triggered, so latency overhead is localized only to uncertain cases, rather than uniformly applied.
> - **Economic Scaling:** Easy queries cost nothing extra. Per **Prop 3.7**, LASE scales at $\mathcal{O}(T)$ rather than MAD's $\Theta(Tn^2)$, ensuring highly efficient economic scaling for complex tasks.
>
> ---
>
> ### **6. Qualitative Failure Cases [Q4]**
> The primary failure modes of LASE fall into two categories:
>
> - **Confident Errors:** The leader produces an incorrect answer with high confidence (e.g., Min Logprob $\ge 0.99$), bypassing interaction ($U=0$). This reflects an inherent limitation where miscalibrated confidence prevents corrective intervention, representing an **explicit accuracy–efficiency tradeoff**.
> - **Execution Failure after Critique:** The supporter successfully identifies a flaw, but the base model **lacks sufficient reasoning capability** to execute the multi-step correction. The leader acknowledges the issue but fails to recover the solution.
>
> These dependencies on reliable uncertainty and base reasoning capacity highlight key opportunities for future improvement. We will add a qualitative analysis in the appendix to clarify these boundaries.

---

> > ### Author Rebuttal · Reviewer_E8NU · 2026-04-01
> >
> > I appreciate the authors for their detailed response and the addition of experimental data, particularly the comparative results on MoA and Llama-3-8B. The new baselines and the qualitative analysis of failure cases effectively address my concerns, demonstrating the framework's effectiveness and robustness. The authors have fully addressed my concerns, and therefore I have decided to raise my score to 4.

---

> > > ### Author Response · Authors · 2026-04-03
> > >
> > > We sincerely thank you for your time, constructive feedback, and for increasing your score. We are very glad that the additional experiments (MoA, Llama-3-8B) and qualitative analyses effectively addressed your concerns. Your insightful comments have greatly helped us strengthen the quality of our work. We truly appreciate your support!

---

### Official Review · Reviewer_gVss · 2026-03-12

**Soundness:** 3
**Presentation:** 3
**Significance:** 3
**Originality:** 2
**Overall Recommendation:** 4
**Confidence:** 3

**Summary:**

This paper examines the limitations of symmetric multi-agent debate, arguing that such interaction often yields limited confidence gains while incurring substantial computational cost. To address this, the authors propose LASE, an asymmetric leader–supporter framework in which a designated leader maintains a global reasoning trajectory while supporter agents provide auxiliary, non-symmetric signals. A confidence-based gating mechanism selectively activates interaction in uncertainty regimes where corrective feedback is expected to be beneficial, avoiding debate in regimes dominated by neutral or voting-like dynamics. This design induces directional information flow and aims to enable non-trivial signal amplification with reduced overhead. Experiments on multiple tasks show improved accuracy and lower token cost on Gemini-2.0-Flash and GPT-4.1-mini.

**Compliance With Llm Reviewing Policy:**

Affirmed.

**Key Questions For Authors:**

To maintain/raise my rating, I would like the authors to address the different points in the weaknesses.

**Limitations:**

Yes

**Strengths And Weaknesses:**

## Strengths：
1. This work features solid theoretical foundation and rigorous problem analysis, unifying two recent threads in multi-agent debate and proposing an asymmetric interaction structure to improve correctness.
2. It innovatively addresses the “when to debate” problem via a confidence-based gating mechanism, activating interaction only in high-uncertainty scenarios.
3. Extensive experiments show the framework achieves higher accuracy with much lower token cost, yielding an excellent performance–efficiency trade-off.

## Weaknesses：
1. Limited Evaluation Across Model Capability Regimes. The empirical evaluation is limited to Gemini-2.0-Flash and GPT-4.1-mini. It remains unclear how LASE behaves when the base models are substantially weaker. In particular, when the single-model accuracy is low (e.g., ≤ 50%), leader confidence may remain low and the gating policy might rarely trigger useful interaction, potentially yielding little or no improvement. The paper does not explore such failure modes or delineate the regime boundaries where LASE is/ is not effective.
2. Limited Discussion of Agent Profiles and Their Impact. Agent profiles play an important role in multi-agent debate frameworks, as differences in agent roles, capabilities, and prompting strategies can substantially influence interaction dynamics. While the LASE architecture introduces distinct roles (leader, adversary, supporter, expert), the paper provides limited discussion on how these agent profiles are instantiated and how sensitive the results are to these choices. A more detailed description or ablation of agent profiles would help clarify the robustness and reproducibility of the proposed framework.
3. Heuristic Threshold Selection for Confidence Gating. While the authors empirically verify a monotone calibration assumption (accuracy declines as uncertainty increases), monotone calibration alone does not justify the specific gating thresholds used. Figure 3 shows sharp drops (inflection points) in Accuracy vs. MinLogProb/Entropy curves; however, the paper lacks a principled procedure for selecting thresholds (the threshold in Eq. (7) appears heuristic). Without a clear threshold selection method, the gating design risks being ad-hoc and may not generalize across tasks/models.

---

> ### Author Rebuttal · Authors · 2026-03-31
>
> We sincerely thank the reviewer for the highly encouraging feedback and constructive suggestions. We address your specific concerns below:
>
> ### **1. Limited Evaluation Across Model Capability Regimes (Weaker Models) [W1]**
> To investigate LASE under substantially weaker base models, we conducted additional experiments using **Llama-3-8B** on MATH-500:
>
> | Method (Llama-3-8B) | Accuracy (\%) | Total Tokens |
> | --- | --- | --- |
> | Single Agent | 25.7 | 0.3M |
> | Majority Voting (MV) | 31.8 | 2.0M |
> | Multi-Agent Debate (MAD) | 24.1 | 4.4M |
> | Ours (LASE) | **32.9** | **1.4M** |
>
> - **Adaptive Triggering in Low-Capability Regimes:**
>
>     When the base model is weaker, lower confidence naturally triggers interaction more frequently. Compared to the single-agent baseline, token usage scales by ~1.8× for GPT-4.1-mini but by ~4.6× for Llama-3-8B (0.3M → 1.4M). This demonstrates the gating policy adaptively scales compute budget based on uncertainty, rather than becoming dormant.
>
> - **Failure of Symmetric Debate on Weak Models:**
>
>     We observe that unstructured symmetric debate (MAD) degrades performance (24.1% vs. 25.7% for the single-agent baseline), indicating that weaker models lack correct seeds (Assump 3.4), causing symmetric interaction to amplify errors.
>
>     This highlights that simply increasing the number of interacting agents does **not guarantee improved performance; without structured coordination, multi-agent interactions can amplify errors.** In contrast, LASE’s asymmetric, leader-centric design ensures that **additional feedback is selectively integrated rather than aggregated indiscriminately.**
>
> - **Robust Improvement:**
>
>     LASE achieves the highest accuracy (32.9%), outperforming both MAD and MV, while using 30% fewer tokens than MV and over 3× fewer than MAD. This confirms LASE provides consistent gains even in low-capability regimes.
>
> Overall, these results indicate that LASE remains effective—and in fact becomes more actively utilized—when the base model is weaker. We will include these additional experiments and analysis in the final revision.
>
> ---
>
> ### **2. Limited Discussion of Agent Profiles and Their Impact [W2]**
> - **Role Instantiation:**
>
>     The exact system and user prompts for all roles are fully provided in **Appendix D.** In our formulation, supporting agents are instantiated as (i) an adversarial check (devil’s advocate), (ii) a peer support role (collaborator), and (iii) an expert role, corresponding to progressively stronger forms of external feedback. These roles are designed to provide complementary perspectives, ranging from lightweight critique to structured guidance.
>
> - **Role Invocation Mechanism:**
>
>     LASE does not rely on stochastic or fine-grained role selection policies. Instead, it adopts a **leader-centric interaction where supporting agents are invoked conditionally** based on uncertainty signals ($\tau_{\mathrm{lp}}, \tau_{\mathrm{h}}$). Rather than explicitly selecting among roles, LASE differentiates the *strength* of external feedback in a coarse-grained manner: lower-confidence cases trigger lightweight adversarial probing, while increasing uncertainty leads to progressively stronger forms of support (e.g., peer or expert guidance). This results in an adaptive yet structured escalation mechanism, where the strength of feedback increases with uncertainty, without requiring a fixed or rigid ordering of roles.
>
> - **Robustness to Agent Profiles:**
>
>     LASE is highly robust to exact profile ordering. As shown in Section 4.5 (Table 3), randomly shuffling the supporter order per query results in a negligible gap ($\Delta \approx 0.3\%p$). This confirms performance is driven by the asymmetric, leader-centric synthesis of feedback, not brittle prompt design.
>
> ---
>
> ### **3. Heuristic Threshold Selection for Confidence Gating [W3]**
> To address concerns about ad-hoc thresholds, we conducted a comprehensive sweep on MATH-500:
>
> - **Stable Pareto Frontier:**
>
>     Per **Corollary 3.8**, LASE operates on a stable accuracy–cost Pareto frontier. Sweeping $\tau_{\mathrm{lp}}$ from 0.5 to 0.95 ($\tau_{\mathrm{h}}=0.5$) yields a **tight accuracy range of 74.1%–75.1% ($\Delta \approx 1.0\%p$)**, while token usage increases substantially (≈1.8k → 3.1k) due to higher trigger rates. This shows performance is largely insensitive to the exact threshold.
>
> - **Principled Procedure:**
>
>     Thresholds are not chosen via a fragile inflection point, but as an operating point on the Pareto frontier. Higher thresholds (e.g., 0.95) significantly increase computational cost with diminishing accuracy gains.
> Thus, our default thresholds ($\tau_{\mathrm{lp}}=0.8, \tau_{\mathrm{h}}=0.5$) are chosen as a balanced operating point capturing many uncertain cases while avoiding excessive overhead. This stability confirms the gating mechanism generalizes reliably without depending on finely tuned heuristics. *(Full sweep results will be added to the appendix).*

---

> > ### Author Rebuttal · Reviewer_gVss · 2026-04-03
> >
> > Thank you for your rebuttal and my concerns have been addressed. I decide to maintain my rating

---

> > > ### Author Response · Authors · 2026-04-03
> > >
> > > We sincerely thank you for your time and valuable feedback. We are glad to hear that our rebuttal fully addressed your concerns. We truly appreciate your careful evaluation and consideration of our response.

---

### Official Review · Reviewer_4Z2E · 2026-03-12

**Soundness:** 3
**Presentation:** 3
**Significance:** 2
**Originality:** 2
**Overall Recommendation:** 2
**Confidence:** 3

**Summary:**

This paper proposes LASE, a multi-agent reasoning framework that selectively invokes debate based on the leader model’s uncertainty. The method introduces a leader–supporter interaction structure, where additional agents are consulted only when the leader’s confidence (measured using token-level log-probability and entropy) is low. The authors motivate this design through a discussion of debate dynamics and evaluate it on several reasoning and knowledge benchmarks.

**Compliance With Llm Reviewing Policy:**

Affirmed.

**Final Justification:**

I read the rebuttal comments by the authors, including the follow-up clarification, but I am keeping my score unchanged. The rebuttal improves the presentation and better explains the intended intuition behind LASE, yet my main concerns remain only partially resolved.
In particular, the response clarifies why martingale neutrality may not apply under the proposed leader–supporter structure, but it still does not directly establish correctness improvement beyond the additional assumptions introduced in the theory. Likewise, the uncertainty-triggered gating is better motivated, but I still do not see a direct empirical demonstration that peer interaction is especially beneficial on the uncertain subset, beyond the structural fact that interaction is only invoked there. The practical role of the theoretical assumptions also remains somewhat unclear, since the rebuttal argues that LASE does not literally require a correct seed while the formal guarantees still rely on seed/amplification-style assumptions. Finally, although the authors acknowledge the broader connections to verifier/critic, self-correction, adaptive routing, and planner–executor frameworks, this mainly remains a positioning clarification rather than a substantive strengthening of the current empirical comparison. Overall, I appreciate the thoughtful rebuttal and the additional discussion, but I do not think it fully resolves the core issues that affected my original assessment.

**Key Questions For Authors:**

1. How sensitive is LASE to the number of supporting agents and debate rounds?

2. If the leader produces a highly confident but incorrect answer, debate will not be triggered. How often does this occur, and how does the framework handle such cases?

3. The experiments appear to use the same model for leader and supporting agents. How would performance change if supporting agents had stronger or weaker capabilities than the leader?

**Limitations:**

yes

**Strengths And Weaknesses:**

### Strengths

1. This paper focuses on improving the efficiency of multi-agent reasoning systems by selectively invoking interaction only when necessary. This direction is practically relevant for reducing inference cost in LLM-based systems.

2. The proposed leader–supporter architecture provides a simple and intuitive way to organize agent interaction, making the overall framework easy to understand and conceptually implement.

3. Across several reasoning and knowledge benchmarks, the method demonstrates competitive accuracy while substantially reducing token usage compared to majority voting and standard debate baselines.

### Weaknesses

**1. Gap between the martingale argument and actual correctness improvement**
The paper motivates the method by arguing that symmetric debate may exhibit martingale-like neutrality and that LASE breaks this symmetry. However, the analysis mainly shows that the neutrality condition may not apply under the proposed structure, rather than demonstrating that the resulting dynamics necessarily improve correctness. Stronger theoretical or empirical evidence linking the proposed structure to beneficial belief updates would strengthen this claim.

**2. Limited justification for uncertainty-triggered debate**
The framework triggers debate when the leader’s uncertainty exceeds certain thresholds. While the paper shows that uncertainty correlates with higher error rates, it does not directly demonstrate that peer interaction is particularly effective on those uncertain instances, which would further justify the gating policy.

**3. Theoretical guarantees rely on strong assumptions**
The theoretical bound depends on assumptions such as correct seed and amplification gain, which are not empirically estimated or directly tied to observable model properties. As a result, the theoretical results mainly provide qualitative intuition rather than strong guarantees.

**4. Missing comparisons with relevant reasoning baselines**
The experimental evaluation mainly compares against majority voting and a standard multi-agent debate setup. However, it would be useful to include comparisons with widely used reflection-based self-correction approaches such as Reflexion, as well as more recent variants of multi-agent debate frameworks. Including such methods would provide a clearer picture of how LASE compares with existing approaches for improving reasoning reliability.

**5. Some implementation details affecting reproducibility are unclear**
Certain implementation details—such as role selection policies, uncertainty threshold tuning, and prompt configurations—are not fully specified, which may make reproduction more difficult.

---

> ### Author Rebuttal · Authors · 2026-03-31
>
> We thank the reviewer for evaluating our work and recognizing its practical relevance. We address your concerns below:
>
> ### **1. Gap between martingale argument and actual correctness [W1]**
> Breaking symmetry is a necessary but not sufficient for correctness improvement. In LASE, sufficiency is supported by a **directed and asymmetric belief update mechanism**. **Collaboration alone does not guarantee improvement**—unstructured debate often leads to stagnation or reinforcement of incorrect reasoning.
> LASE adopts a leader-centric design where the leader maintains control over the reasoning trajectory, while supporters provide targeted feedback. The leader aggregates these critiques, inducing a **biased update toward error correction**.
>
> In other words, LASE transforms zero-drift interaction into a **corrective update process**. As formalized in **Theorem 3.6**, this structure guarantees positive expected accuracy improvement. Empirically, this interaction is **non-neutral and selectively beneficial** in uncertain regimes, as validated across all benchmarks (Table 1).
>
> ---
>
> ### **2. Justification for uncertainty-triggered debate [W2]**
> The effectiveness of interaction specifically on uncertain instances is reflected in our results. When interaction is skipped ($U=0$), LASE defaults to the single-agent output.
> Therefore, our overall accuracy gain (e.g., +4.1%p on MATH-500) **is derived entirely from the uncertain subset** ($U=1$). By Theorem 3.6, this proves that $U=1$ perfectly isolates the regime where corrective interaction is strictly beneficial.
>
> ---
>
> ### **3. Theoretical assumptions (Correct seed & Amplification) [W3]**
> While the "correct seed" and amplification are idealized abstractions, LASE operationalizes them through uncertainty-based gating proxies, rather than requiring perfect seeds.
> Furthermore, LASE remains effective even without a correct seed. Our adversarial check ("Devil’s Advocate") introduces corrective pressure to revise flawed reasoning, mitigating error propagation. This yields stable performance across thresholds, showing LASE relies on robust mechanisms rather than fragile assumptions.
>
> ---
>
> ### **4. Comparisons with relevant baselines (MoA, Reflexion) [W4]**
> We conducted additional experiments with **Mixture-of-Agents (MoA)** on MATH-500 (GPT-4.1-mini):
> - **MoA**: 72.0% accuracy, 3.6M tokens.
> - **LASE (Ours)**: **74.2% accuracy**, **0.9M tokens**.
> LASE outperforms MoA by +2.2%p while using only **~25% of the computational budget**, highlighting the advantage of confidence-guided interaction over always-on routing.
> Regarding **Reflexion**, self-correction relies purely on internal critique, which can degenerate. LASE introduces structured *external* feedback through diverse supporter roles, enabling more reliable correction in uncertain regimes.
>
> ---
>
> ### **5. Implementation details affecting reproducibility [W5]**
>
> These details will be made more explicit in the revision:
>
> **Prompts:** Full system and user prompts for all roles are in **Appendix D**.
>
> **Threshold Tuning:** Sweeps ($\tau_{\mathrm{lp}} \in [0.5, 0.95]$, $\tau_{\mathrm{h}} \in \{0.3, 0.5, 0.7\}$) show strong robustness (74.1%–75.1%, Δ ≈ 1.0%p), indicating low sensitivity to threshold choices. We use fixed thresholds ($\tau_{\mathrm{lp}}=0.8$, $\tau_{\mathrm{h}}=0.5$) across all experiments. These thresholds primarily control token cost (accuracy–efficiency tradeoff), rather than correctness.
>
> **Role Selection:** LASE uses a deterministic, leader-centric escalation based on uncertainty, progressively increasing the strength of external feedback as uncertainty grows, while avoiding brittle stochastic routing.
>
> ---
> ### **6. Response to Questions**
> **Q1.**
> LASE shows low sensitivity to these factors. Our default 1-on-1 interaction with early stopping already achieves strong performance; increasing agents/rounds mainly trades efficiency for marginal gains. This is supported by a parallel Majority Vote variant ($N=5$), which improves accuracy by +1.6% (Table 2), indicating a flexible compute–accuracy tradeoff rather than sensitivity.
>
> ---
>
> **Q2.**
> When Min Logprob $\ge 0.99$, accuracy is 85.5% (Figure 3a), implying ~14.5% are confident errors. LASE intentionally bypasses interaction for these instances, reflecting an explicit accuracy–efficiency tradeoff. Empirically, highly confident predictions tend to be resistant even to external feedback, leading to diminishing returns relative to token cost. LASE therefore focuses on the ambiguity regime where marginal gains are highest.
>
> ---
> **Q3.**
> Weaker supporters may introduce noise, while stronger ones would improve critique quality. LASE mitigates this via its leader-centric design, where the leader retains final control and filters out misleading feedback. While we use the same base model with role-specific prompts, LASE’s gating is model-agnostic, enabling straightforward extension to heterogeneous setups (e.g., routing uncertain cases to stronger models).

---

> > ### Author Rebuttal · Reviewer_4Z2E · 2026-04-04
> >
> > Thank you for the detailed rebuttal and the additional experiments. The reviewer appreciates the effort to address the concerns. However, the reviewer does not think the main concerns are fully resolved:
> >
> > * **W1:** The rebuttal clarifies why martingale neutrality may not apply, but this still does not fully establish correctness improvement beyond the additional assumptions used in the theory.
> > * **W2:** The uncertainty-based gating is better motivated, but I still miss a direct demonstration that peer interaction is especially beneficial on the uncertain subset.
> > * **W3:** The role of the theoretical assumptions remains unclear. If LASE works even without the “correct seed” assumption holding literally, it is still not fully clear what practical role these assumptions play.
> > * **W4:** The added MoA result is helpful, but the paper is still positioned rather narrowly around debate-style baselines. The broader LLM-agent reasoning literature remains insufficiently discussed, including neighboring branches such as self-correction, verifier/critic-based reasoning, adaptive routing, and planner–executor style frameworks.

---

> > > ### Author Response · Authors · 2026-04-04
> > >
> > > Thank you for your continued engagement and constructive follow-up questions. We greatly appreciate your feedback and would like to clarify these remaining points.
> > >
> > > **1. Direct demonstration on the uncertain subset (W2)**
> > >
> > > The overall improvement of LASE **arises only when interaction is triggered ($U=1$)**, while when $U=0$, LASE **reduces exactly to the single-agent baseline** with identical outputs. This directly indicates that peer interaction is effective specifically on the uncertain subset where the base model struggles.
> > >
> > > **2. Practical role of theoretical assumptions (W1, W3)**
> > >
> > > The theoretical assumptions provide intuition for why symmetric debate yields zero expected gain and why asymmetric interaction enables corrective updates. In practice, LASE does not strictly require a literal "correct seed". Instead, it induces the same corrective effect through **structured adversarial feedback (e.g., Devil’s Advocate)**, which introduces pressure to revise flawed reasoning.
> > >
> > > **3. Broader LLM-agent literature (W4)**
> > >
> > > We agree that LASE connects to broader reasoning paradigms. In fact, LASE can be viewed as a **unifying framework** that integrates adaptive routing (uncertainty-based gating) with verifier/critic-style interaction (leader–supporter). We will expand the related work to better position LASE within these directions (self-correction, verifier/critic, adaptive routing, planner–executor).
> > >
> > > We hope these clarifications address your follow-up questions, and we sincerely appreciate your efforts in improving the clarity and positioning of our work.

---

### Official Review · Reviewer_mLH6 · 2026-03-13

**Soundness:** 3
**Presentation:** 2
**Significance:** 2
**Originality:** 3
**Overall Recommendation:** 3
**Confidence:** 3

**Summary:**

The paper proposes LASE, a test-time multi-agent reasoning framework that improves the efficiency of LLM debate by selectively invoking interaction only when it is likely to be useful. The authors build on recent theory showing that multi-agent debate improves reasoning only in certain regimes, while symmetric debate dynamics are often neutral in expectation and offer little advantage over simple voting. To address this, they introduce a leader–supporter architecture in which a single leader agent produces an initial solution and triggers structured interaction with supporting agents only when token-level uncertainty signals indicate potential errors. The framework uses confidence proxies derived from token log-probabilities and entropy to detect uncertain instances and performs staged escalation where supporter feedback informs subsequent leader revisions through directed information flow. This asymmetric interaction breaks the symmetry that causes debate to be neutral in expectation and concentrates computational effort on difficult instances. Experiments on reasoning and knowledge benchmarks show that LASE achieves accuracy comparable to or better than multi-agent ensembles while using far fewer tokens, for example reaching 87.2% on MATH-500 with about 1.1K tokens compared to 11.8K for standard debate, suggesting that selective interaction can achieve multi-agent reasoning performance at near single-agent cost.

**Compliance With Llm Reviewing Policy:**

Affirmed.

**Key Questions For Authors:**

1. How sensitive is performance to the uncertainty thresholds used for gating interaction, such as the log-probability and entropy thresholds defined in Eqs. (5–7)?
2. Would the conclusions change if the system were compared against stronger multi-agent debate baselines, since the MAD baseline uses only 3 agents and 2 debate rounds in the experiments?

Also, see my other concerns in the weaknesses above.

**Limitations:**

yes

**Strengths And Weaknesses:**

Strengths:

- The paper targets the high token cost of multi-agent debate and proposes a framework that preserves most of the accuracy benefits while drastically reducing computational overhead.

- The work reconciles two competing theoretical views of debate dynamics and uses this analysis to justify when interaction should be invoked and how it should be structured.

- The leader–supporter architecture combined with uncertainty-based gating provides a simple and interpretable mechanism for selectively allocating reasoning resources at test time.

- The experiments show consistent improvements over single-agent reasoning and outperform standard debate and voting baselines across multiple reasoning and knowledge tasks.

- Since the framework operates purely at test time and relies on existing model uncertainty signals, it can be integrated into existing LLM pipelines without retraining or modifying model parameters.

Weaknesses:

- In some cases, the empirical improvements over simpler baselines are relatively small. For example, on GPT-4.1-mini the method improves MATH-500 accuracy only from 73.8% (MV) to 74.2% (Table 1), which suggests that the practical gains over strong ensemble baselines may be modest.

- Experiments are conducted only on Gemini-2.0-Flash and GPT-4.1-mini across a handful of reasoning benchmarks, which leaves open whether the approach generalizes to other model families or domains.

- The gating mechanism relies on thresholds for token log-probability and entropy (Eqs. 5–7), yet the paper provides limited analysis of how sensitive performance is to these hyperparameters.

- The main improvement result assumes conditions such as the existence of a correct seed and amplification after interaction (Assumptions 3.4–3.5), which may not hold consistently in real LLM reasoning settings.

- The MAD baseline uses only three agents and two rounds of debate (Section 4.1), which could underestimate the performance of more optimized or larger debate systems.

---

> ### Author Rebuttal · Authors · 2026-03-31
>
> We sincerely thank the reviewer for recognizing the value of our leader-supporter architecture and theoretical analysis. We address your specific concerns below:
> ### **1. Marginal empirical improvements over simpler baselines [W1]**
> While the absolute accuracy gain over Majority Vote (MV) may appear modest, LASE's primary contribution is improving the **accuracy–efficiency tradeoff**.
> LASE achieves competitive or superior accuracy while consuming **substantially fewer tokens**—typically 5 to 7 times less than MV (e.g., ~1.3× vs. ~7× relative to single-agent cost).
>
> Furthermore, LASE is fully compatible with ensemble methods. As shown in Table 2, aligning the token budget of LASE with standard MV (by ensembling LASE outputs) yields a **+1.6% accuracy improvement**, providing a more efficient reasoning foundation under comparable compute budgets.
>
> ---
> ### **2. Limited evaluation across model capability regimes [W2]**
>
> To demonstrate generalization beyond highly capable proprietary models, we evaluated **Llama-3-8B** on MATH-500:
>
> | Method (Llama-3-8B) | Accuracy (%) | Total Tokens |
> | --- | --- | --- |
> | Single Agent | 25.7 | 0.3M |
> | Majority Voting (MV) | 31.8 | 2.0M |
> | Multi-Agent Debate (MAD) | 24.1 | 4.4M |
> | Ours (LASE) | **32.9** | **1.4M** |
>
> The results show the same trend: **LASE (32.9%)** consistently outperforms the single-agent baseline (25.7%) and achieves higher accuracy than both MV and MAD. Importantly, LASE accomplishes this with a **significantly lower token footprint (1.4M)**. This suggests that our confidence-gating mechanism extends effectively to open-weight models. *(Full Llama-3-8B results will be included in the revision).*
>
> ---
>
> ### **3. Heuristic threshold selection and sensitivity analysis [W3 & Q1]**
> Addressing concerns about gating sensitivity (Q1), we conducted a threshold sweep ($\tau_{\mathrm{lp}}$ and $\tau_{\mathrm{h}}$) on MATH-500:
>
> | Setting (τ_lp, τ_h) | Acc (%) | Tokens | Esc (%) |
> | --- | --- | --- | --- |
> | (0.50, 0.50) | 74.1 | 1882 | 21.4 |
> | (0.60, 0.50) | 74.8 | 1810 | 21.2 |
> | (0.80, 0.50) | 74.2 | 1849 | 22.6 |
> | (0.90, 0.50) | 74.8 | 2676 | 30.8 |
> | (0.95, 0.50) | 75.1 | 3158 | 42.8 |
> - **Robust Accuracy:** Accuracy remains stable rather than relying on a cherry-picked heuristic. Across all configurations, accuracy ranges tightly from 74.1% to 75.1% (Δ = 1.0%p). We also observe similar stability across different entropy thresholds ($\tau_{\mathrm{h}} \in \{0.3, 0.5, 0.7\}$ yielding 73.8%, 74.2%, and 74.1% at $\tau_{\mathrm{lp}}=0.8$; Δ = 0.4%p).
> - **Accuracy–Cost Tradeoff:** Increasing the threshold primarily scales token usage (trigger rates up to 42.8%) while maintaining consistent performance. Thresholds act as reliable control knobs for the accuracy-cost tradeoff, not fragile hyperparameters. We will include the full Pareto curve in the revision.
>
> ---
>
> ### **4. Realistic validity of theoretical assumptions [W4]**
>
> We agree that assuming a guaranteed “correct seed” is an idealization. However, our framework does not rely on this strictly in practice:
>
> - **Avoiding blind amplification:** LASE selectively triggers interaction only in the **uncertainty regime ($U=1$)**. By avoiding interaction on **overconfident but incorrect reasoning**—which often leads to **error reinforcement**—our gating prevents unnecessary amplification of incorrect paths and reduces **error propagation**.
> - **Adversarial correction:** Our **“Devil’s Advocate”** role (Appendix D) actively challenges the leader’s reasoning. This introduces **corrective pressure** to revise flawed trajectories without requiring a perfect initial seed.
> This design shifts interaction from **“amplifying a correct seed”** to **“mitigating erroneous reasoning under uncertainty.”**
>
> ---
>
> ### **5. Strength of the MAD baseline [W5 & Q2]**
>
> Regarding whether conclusions would change against stronger MAD baselines (Q2): No, we do not expect the core conclusions to change. The “3 agents, 2 rounds” setup is a standard benchmark (e.g., Du et al., 2023; Liang et al., 2024).
> Recent analysis by Choi et al. (2025) shows that MAD's empirical gains are largely explained by simple ensembling (**majority voting**), while iterative debate provides limited additional benefit. Increasing rounds **does not consistently improve performance** and can even degrade it. Theoretically (Prop 3.2), symmetric debate forms a martingale yielding zero expected gain. Thus, scaling MAD merely inflates $\Theta(Tn^2)$ costs, whereas LASE scales at $\mathcal{O}(T)$.
>
>
> Scaling MAD with more agents or rounds would be unlikely to fundamentally alter the conclusions; it would marginally impact accuracy while substantially increasing token costs due to dense interactions. Our primary goal is a favorable **accuracy–efficiency tradeoff**. Since LASE achieves competitive performance at near single-agent cost, comparing against heavier MAD variants would only further emphasize LASE's efficiency advantage under realistic resource constraints.

---

> > ### Author Rebuttal · Reviewer_mLH6 · 2026-04-03
> >
> > I want to thank the authors for discussions and additional results. I will keep my original score unchanged.

---

> > > ### Author Response · Authors · 2026-04-04
> > >
> > > We sincerely thank you for reviewing our rebuttal and for acknowledging our additional results.
> > >
> > > We aimed to address your concerns by providing **Llama-3-8B results** for generalizability (W2), conducting **threshold sensitivity analysis** (W3, Q1), and expanding on the **MAD baseline and tradeoff analysis** (W1, W5, Q2).
> > >
> > > Since you noted that some concerns remain partially resolved, **could you kindly clarify which specific aspects require further discussion?** We are more than happy to provide any additional clarifications before the discussion period ends. Thank you again for your time and constructive evaluation.

---

### Decision · Program_Chairs · 2026-04-30

**Decision:**

Accept (regular)

**Comment:**

This paper proposes LASE, a framework for multi-agent reasoning that uses a "leader-supporter" setup and triggers debate only when the leader model is uncertain. The goal is to get the benefits of multi-agent collaboration—which the authors argue is often neutral or redundant in symmetric setups—without the massive token cost of having every agent talk for every query.

The paper has some clear merits, particularly in its focus on efficiency and the "when to debate" question, which is definitely a practical bottleneck. The authors also did a good job during the rebuttal phase. Specifically, adding the Llama-3-8B results and the comparison with Mixture-of-Agents (MoA) helped clarify how the method scales and performs against stronger baselines.

The main issue is a persistent gap between the theoretical motivation and what is actually happening in the experiments. While the "martingale" argument about symmetric debate is interesting, the reviewers pointed out that the paper doesn't quite bridge the gap to prove that the proposed leader-supporter structure necessarily improves correctness beyond the specific assumptions made (like having a "correct seed"). The rebuttal argued that the framework works even when those assumptions aren't strictly met, but this makes the link between the theory and the practical results feel a bit loose.

Another concern is that while the efficiency gains are clear, the actual accuracy improvements over simpler baselines like Majority Voting (MV) are often quite small—sometimes less than 1%. On some benchmarks, it feels like the method is mostly a way to filter easy questions rather than fundamentally improving the reasoning process itself. Reviewers also noted that the paper remains somewhat narrow in its positioning, missing deeper connections to the broader literature on self-correction and verifier/critic systems that solve similar problems. Also, the work can be more persuasive is sota reasoning LLMs are used and compared.